# An artificial visual neuron with multiplexed rate and time-to-first-spike coding

Fanfan Li[1,2,7], Dingwei Li[2,7], Chuanqing Wang[3,7], Guolei Liu[2], Rui Wang[4], Huihui Ren[2], Yingjie Tang [2], Yan Wang [2], Yitong Chen[2], Kun Liang[2], Qi Huang[5], Mohamad Sawan[3,5,6], Min Qiu [2,5,6], Hong Wang [4] ✉ & Bowen Zhu [2,5,6] ✉

Human visual neurons rely on event-driven, energy-efficient spikes for communication, while silicon image sensors do not. The energy-budget mismatch between biological systems and machine vision technology has inspired the development of artificial visual neurons for use in spiking neural network (SNN). However, the lack of multiplexed data coding schemes reduces the ability of artificial visual neurons in SNN to emulate the visual perception ability of biological systems. Here, we present an artificial visual spiking neuron that enables rate and temporal fusion (RTF) coding of external visual information. The artificial neuron can code visual information at different spiking frequencies (rate coding) and enables precise and energy-efficient time-to-first-spike (TTFS) coding. This multiplexed sensory coding scheme could improve the computing capability and efficacy of artificial visual neurons. A hardware-based SNN with the RTF coding scheme exhibits good consistency with real-world ground truth data and achieves highly accurate steering and speed predictions for self-driving vehicles in complex conditions. The multiplexed RTF coding scheme demonstrates the feasibility of developing highly efficient spike-based neuromorphic hardware.

The human eye can quickly and efficiently perceive visual information in complex environments, including target feature extraction and classification[1-5]. In human visual sensory transduction, sensory neurons encode visual stimulus information into event-driven neural spikes and transmit this information to the brain for perception. These neural spikes are encoded with multiplexed spatial-temporal information, representing individual visual stimulus variables with rich features and high efficacy[6-9]. For example, stronger light stimuli could lead to both a higher firing frequency (rate coding) and a shorter latency for the first spike after the stimulus (time-to-first-spike, TTFS)[10,11]. Rate coding is a basic neural coding mechanism in which retinal stimuli are encoded based on the number of spikes that occur during a certain encoding window; however, rate coding cannot provide efficient temporal information or sufficient features to fully represent the stimulus[12]. On the other hand, TTFS coding (latency coding) is a fast temporal encoding method with robustness to noise and the highest efficiency in terms of spike counts, in which the stimulus onset is precisely 'locked' to the first spike time[4]. Thus, TTFS coding is superior to rate coding and more reliable in urgent situations, such as obstacle avoidance and threat or ally recognition[13]. With complementary rate and TTFS coding, the natural visual system can efficiently process complex visual information within 150 ms[14].

In comparison, complementary metal–oxide–semiconductor (CMOS) image sensors work according to a frame-driven approach

[1]School of Materials Science and Engineering, Zhejiang University, Hangzhou, China. [2]Key Laboratory of 3D Micro/Nano Fabrication and Characterization of Zhejiang Province, School of Engineering, Westlake University, Hangzhou, China. [3]CenBRAIN Neurotech, School of Engineering, Westlake University, Hangzhou, China. [4]Key Laboratory of Wide Band Gap Semiconductor Technology, School of Microelectronics, Xidian University, Xi'an, China. [5]Westlake Institute for Optoelectronics, Westlake University, Hangzhou, China. [6]Institute of Advanced Technology, Westlake Institute for Advanced Study, Hangzhou, China. [7]These authors contributed equally: Fanfan Li, Dingwei Li, Chuanqing Wang. ✉e-mail: hongwang@xidian.edu.cn; zhubowen@westlake.edu.cn

with high energy budgets[15]. The large mismatch between natural and machine visual efficacies has inspired the development of artificial visual neurons, which can encode visual information into binary spike trains and be implemented in spiking neural networks (SNN) to process visual data with high efficiency and high biological fidelity[16-19]. Rate coding is commonly used in SNN to represent the intensity of external stimuli; however, this approach cannot provide sufficient temporal information, and the processing speed is limited by the average firing rate[20-23]. In contrast, TTFS coding, which resembles natural vision, can provide important spatiotemporal information for the implementation of SNN to process dynamic visual data with high sparsity and high energy efficiency. Hardware-based SNN with TTFS coding is more efficient than SNN with rate coding, demonstrating faster speed and reduced energy consumption[24-27]. Although precise temporal encoding has been achieved in artificial visual neuron systems, fusing rate and TTFS coding in a single spike train has not yet been realised in SNN hardware, compromising the capacity of such networks to rapidly and accurately process visual data in complex visual environments[28-35].

Here, we report a biomimetic artificial spiking visual neuron that can encode analogue visual stimuli into relevant spike trains with both rate and TTFS coding. The artificial neuron integrates an $In_2O_3$ synaptic phototransistor and an $NbO_x$ Mott memristor, which resemble biological photoreceptors and retinal ganglion neurons, respectively. With the integration of rate and TTFS coding, the biologically plausible artificial neuron can emulate natural vision. Stronger light intensity leads to incremental firing rates (from 0.35 to 1.85 MHz) and shortened first-spike arrival times (or spike latency, from 13.00 to 1.04 µs), outperforming biological counterparts in terms of spiking frequencies (0–100 Hz) and first spike latencies ($\geq 10$ ms). High-frequency event spikes can convey more information in a shorter time, allowing the system to quickly make decisions and execute responses. Moreover, with energy-efficient TTFS coding, the artificial visual neuron can rapidly and precisely detect and encode temporal changes in external stimuli, which is useful in applications requiring high temporal resolution. Furthermore, the artificial neuron is small (the Mott memristor has an active area of $7 \times 7\,\mu m^2$), durable ($10^{10}$ operating cycles), and consumes a low energy of less than 1.06 nJ per spike to accomplish complementary rate and TTFS coding without the need for auxiliary reset circuits. Finally, we implement complementary rate and TTFS coding in a trained SNN, which provides more channels for information to be transferred and further improves SNN efficiency. The SNN can predict the speed and steering angle of autonomous vehicles in complex environments with a low loss function of < 0.5. Our results prove that SNN with the proposed rate-temporal fusion (RTF) encoding scheme can enhance the efficacy of artificial vision with a biologically plausible approach.

## Results

### Biomimetic signal encoding with artificial spiking visual neurons

An overview of the artificial spiking visual neuron capable of both rate and TTFS coding is shown in Fig. 1. Figure 1a shows a schematic of a retina composed of photoreceptors, synapses, and neurons. In natural visual transduction, photoreceptors detect external optical signals and transform them into graded potentials. These potentials influence synaptic plasticity and thus play critical roles in learning and memory. Subsequently, retinal ganglion cells, acting as neurons, encode the processed graded potentials from synapses as electrical spikes, which are action potentials that convey information to the brain for further processing. As shown in Fig. 1b, in rate coding, stronger stimuli lead to higher firing frequencies in an encoding window and vice versa. Moreover, stronger light stimuli lead to shorter latencies for the first neuron spikes and vice versa (TTFS coding). Furthermore, increasing the input synaptic weights can increase the membrane potential, allowing neurons to rapidly reach or exceed the threshold for spike

firing. Thus, neurons fire at higher frequencies and emit their first spikes more quickly.

To emulate the visual phototransduction process, our artificial spiking visual neuron device is composed of an $In_2O_3$ optoelectronic synaptic transistor and an $NbO_x$ Mott spiking neuron (Fig. 1c). As shown in the schematic (Fig. 1d), the $In_2O_3$ synaptic transistor enables optoelectronic synaptic plasticity, and the $NbO_x$ Mott neuron encodes the received optoelectronic signals as stimuli-related electrical spikes via multiplexed RTF coding. The spiking visual neuron is activated when the excitatory postsynaptic potential reaches a threshold. With the RTF encoding scheme, the artificial neuron can rapidly and efficiently represent rich stimulus characteristics.

### Electrical characteristics of artificial neurons and optoelectronic synapses

Threshold switching (TS) $NbO_x$ Mott memristors based on insulator-to-metal transitions (IMTs) can emulate the high-order neural dynamics of biological neurons[36-39]. The fabricated $NbO_x$ memristor had a crossbar structure with an active area of $7 \times 7\,\mu m^2$ (see "Methods" for the details of the fabrication processes). A cross-sectional transmission electron microscopy (TEM) image of the device is presented in Fig. 2a. The stacked layer-by-layer Pt/Ti/$NbO_x$/Pt/Ti films were confirmed. Notably, a nanoscale crystalline region is observed in the high-resolution TEM (HRTEM) image after the initial formation process, corresponding to the $NbO_2$ tetragonal structure (Supplementary Fig. 1). The $NbO_x$ layer in the pristine film is amorphous, and its stoichiometry was determined by X-ray photoelectron spectroscopy (XPS) (Supplementary Fig. 2). The formed $NbO_x$ memristors exhibit volatile TS characteristics (Supplementary Fig. 3).

Figure 2b shows the typical current-voltage (I-V) curves of the $NbO_x$ memristor with TS characteristics (Fig. 2b). The device exhibits a transition from a high resistance state (HRS) to low resistance state (LRS) when the applied voltage surpasses a threshold voltage ($V_{th}$) of -1.37 V. Conversely, the device returns to its HRS when the applied voltage is less than the holding voltage ($V_{hold}$) of -1.17 V. In addition, $R_{LRS}$ is 293.3 Ω, and $R_{HRS}$ is 5349.5 Ω. Current compliance ($I_{CC}$) of 4 mA was applied to prevent permanent breakdown. The volatile resistive switching of $NbO_2$ occurs due to the IMT Mott transition. Correspondingly, an "S"-shaped negative differential resistance (NDR) behaviour is observed when sourcing with a current sweep (Supplementary Fig. 4), which occurs due to the thermally induced changes in conductivity[36]. The Mott device exhibited the best endurance performance among various volatile TS nanodevices, and the device could operate for more than $10^{10}$ cycles driven by consecutive electrical pulses (Fig. 2c). Pulse operation with the endurance characteristics of the $NbO_x$ Mott memristors is further illustrated (Supplementary Fig. 5). In addition, the Mott transition enables the device to have a fast-switching speed, needing < 40 ns to switch from the off state to the on state and < 50 ns to return to the off state (Supplementary Fig. 6). The extracted coefficient of variation ($C_v$) values of different parameters ($V_{th}$, $V_{hold}$, and $V_{forming}$) of 100 $NbO_x$ Mott memristors are 0.0349, 0.0303, and 0.0233, respectively, demonstrating low device-to-device variability (Supplementary Fig. 7).

The optoelectronic synaptic transistors have a bottom gate, top contact (BGTC) configurations with solution-processed $In_2O_3$ as the active channel[40]. A cross-sectional HRTEM image reveals the presence of an ~5-nm-thick amorphous $In_2O_3$ channel layer (Fig. 2d). The corresponding device-to-device variation in mobility ($\mu_{sat}$), threshold voltage ($V_{th}$), subthreshold swing (SS), and current on/off ratio ($I_{on/off}$) of 100 $In_2O_3$ phototransistors with values of 0.19, 0.36, 0.21, and 0.52, respectively demonstrate a low variability (Supplementary Fig. 7). The $In_2O_3$ film exhibits strong ultraviolet light absorption (< 400 nm), as shown in the ultraviolet–visible (UV–vis) absorbance spectrum (Supplementary Fig. 8). Figure 2e depicts the transfer curves of the $In_2O_3$ phototransistor under UV radiation with a wavelength of 365 nm at

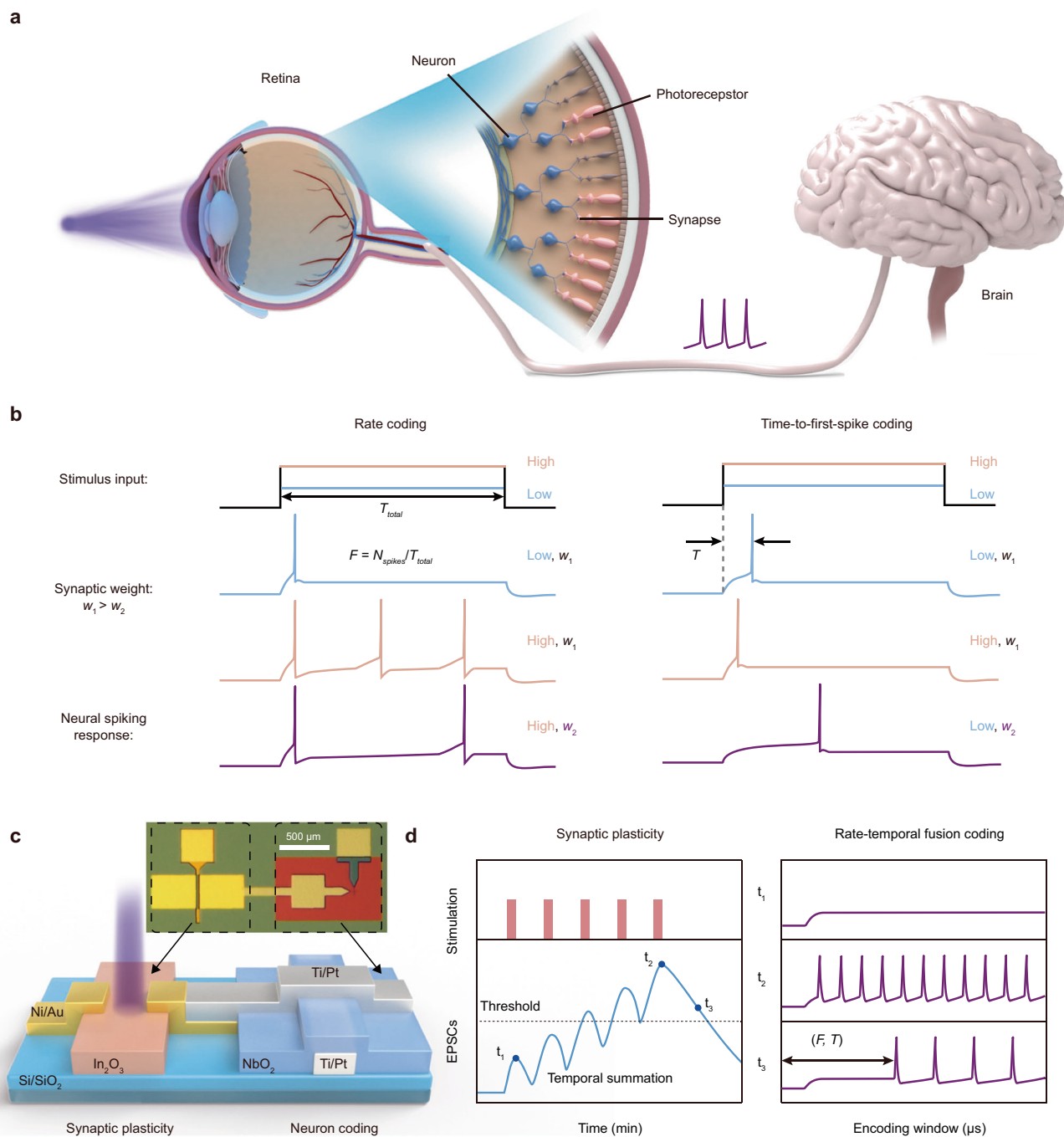

**Fig. 1 | Biomimetic signal encoding with an artificial spiking visual neuron.**
**a** Schematic illustration of the retina, which is composed of photoreceptors, synapses, and neurons. Photoreceptors can respond to external optical signals and convert them into graded potentials. In synapses, synaptic plasticity is responsible for learning and storing memories. Neurons (retinal ganglion cells) can encode synapse-processed graded potentials as action potentials (electrical spikes) to be processed by the brain. **b** Encoding of different input stimuli and synaptic weights by time-to-first-spike (TTFS) coding and rate coding schemes in a biological spiking visual neuron. The frequency ($F$) of rate coding depends on the number of spikes ($N_{spikes}$) within the time window ($T_{total}$), while TTFS coding depends on the first

spike latency ($T$). Low stimulus input (orange) and high stimulus input (blue) along with synaptic weights $w_1$ (black) and $w_2$ (purple) correspond to neural spiking responses. **c** Schematic and an optical image of artificial neuron device composed of the integrated $In_2O_3$ optoelectronic synaptic transistor and $NbO_x$ Mott neuron (1T1R). **d** The optoelectronic retina enables synaptic plasticity and rate-temporal fusion coding. Spiking sensory neurons are activated when EPSPs reach a certain threshold. The rate-temporal fusion encoding of spiking neurons represents the characteristics of the stimulus in real-time through the change in the first spike latency and the spike frequency.

varying power densities ranging from 1.57 to 3.72 mW/cm², illustrating significantly increased channel conductance due to the distinct UV photoresponse of the oxide semiconductor. With these intrinsic optoelectronic properties, the $In_2O_3$ transistor can feasibly be applied as an optoelectronic synapse. As shown in Fig. 2f, the device was illuminated under UV light (365 nm, 1.71 mW/cm², 5 ms, 50 cycles), and a

voltage bias ($V_{DS} = 3$ V) was applied to read the change in the drain current, which corresponds to the excitatory postsynaptic current (EPSC). The EPSC decreased slowly over time when UV light input ceased due to the persistent photoconductivity (PPC) phenomenon[41]. With increased input UV pulses, the synaptic phototransistor showed typical long-term plasticity (LTP) behaviour. In addition, paired-pulse

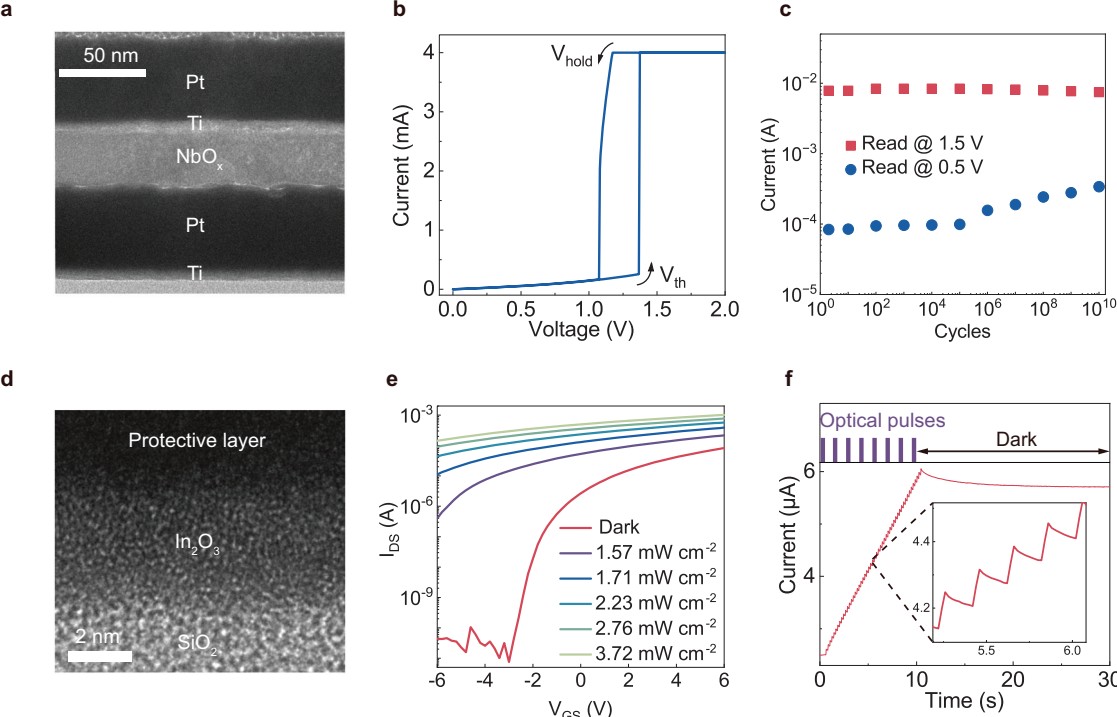

**Fig. 2 | Device characteristics of the NbO$_x$ neurons and In$_2$O$_3$ optoelectronic synaptic transistors. a** Cross-sectional TEM image of a NbO$_x$ Mott neuron. **b** Current-voltage characteristics of a NbO$_x$ Mott neuron. **c** Endurance characteristics of the NbO$_x$ Mott neuron, which can fire stably for more than $10^{10}$ cycles. Transient electrical measurements show that TS is triggered by a voltage pulse with a width of 1 μs and an amplitude of 1.60 V. **d** Cross-sectional TEM image of In$_2$O$_3$ optoelectronic synaptic transistors. **e** Transfer curves of the In$_2$O$_3$ synapse as a function of light power density (from 1.57 to 3.72 mW/cm$^2$, $\lambda$ = 365 nm). I$_{DS}$ versus V$_G$ measured at a drain bias of V$_D$ = 3 V. **f** EPSCs (red line) of the In$_2$O$_3$ synaptic TFT triggered by optical pulses (purple) ($\lambda$ = 365 nm, 5 ms, 5 Hz).

facilitation (PPF), a behaviour related to short-term plasticity (STP), was characterised by applying a pair of optical pulses (Supplementary Fig. 9). The synaptic behaviours of the In$_2$O$_3$ transistors demonstrate their applicability as optoelectronic synapses in artificial visual neural networks.

### Demonstration of rate and TTFS coding

In the natural visual neural system, neurons appear to represent recognised features when they emit spikes[6]. Spike trains (action potentials) carry information about the average firing rate and spike time[10]. Figure 3a illustrates the major differences among rate coding, TTFS coding, and RTF coding. In rate coding, the intensity of an external stimulus is represented by the average spiking rate within a sampling window. However, rate coding has a limited range of stimulus changes, a long processing period, and slow information transmission. In TTFS coding, a neuron encodes its real-valued stimulus-induced response as its spike latency in response to that stimulus. This single-spike coding scheme enables fast and sparse information processing, and it enhances sensitivity to minor variations in input. Thus, multiplexed neural coding schemes operating on different timescales can encode complementary stimulus features, meeting the physiological constraints and reaction times observed in humans and animals, thereby enhancing the capacity of the coding scheme.

To emulate the multiplexed coding scheme observed in the natural visual system, we implement RTF coding in our artificial visual neural system by utilising the spike rate and TTFS to encode visual information. This fast and precise coding scheme could enable a biologically plausible neuromorphic hardware system with high accuracy and fast emulation of SNN. As described earlier, the NbO$_x$ memristor and In$_2$O$_3$ phototransistor have time-dependent neuronal and synaptic functionalities, respectively. The In$_2$O$_3$ phototransistor integrated in

series with a two-terminal NbO$_x$ memristor results in a 1-transistor-1-memristor (1T1R) configuration that can fully represent the functionality of the visual neural pathway in the retina for data encoding and processing.

The circuit schematic of the 1T1R artificial spiking visual neuron is shown in Fig. 3b. An optical laser was used to provide light stimuli (365 nm) for the In$_2$O$_3$ phototransistor. After the light stimuli was ceased, a single electrical pulse (V$_{DD}$ = 3 V, 20 μs width) was applied to the drain of the phototransistor under a gate bias voltage (V$_G$ = 5 V) to measure the encoded current pulses. The optical pulses and the drain current (I$_D$) were regarded as the light stimulus input and artificial neural signal output, respectively. Due to the TS characteristics of NbO$_x$, self-sustained spiking can be obtained in a range of R$_{LRS}$≪R$_{ch}$≪R$_{HRS}$, where R$_{ch}$ is the channel resistance of the In$_2$O$_3$ transistor and R$_{HRS}$ and R$_{LRS}$ are the insulating and metallic resistances of the NbO$_x$ memristor, respectively[41]. When V$_{DD}$ and V$_G$ are fixed, R$_{ch}$ is determined by the parameters of the optical pulses (Supplementary Fig. 10). The spiking behaviours of the artificial neuron can be altered by adjusting the R$_{ch}$ values based on a leaky integrate-and-fire (LIF) model, leading to different spiking durations $\tau_{integration}$ (~R$_{ch}$C) (Supplementary Fig. 11). With this approach, we connect the neural coding behaviour of the artificial visual neuron to the R$_{ch}$ value of the series transistor, which is configured by visual stimuli[42].

Then, we measured the spike patterns generated by the spiking neuron as a function of light intensity and pulse number. Before light pulse illumination, initial light exposure was applied (1.57–3.72 mW/cm$^2$, 10 s) to adjust the baseline current value to ~405 μA and the R$_{ch}$ value to a reference value matching the TS characteristics of the NbO$_x$ memristor. After each test, an electrical pulse (V$_G$: 10 V, 20 μs) is applied to the gate electrode of the phototransistor to initialise its state. This process ensures the restoration of ionised oxygen vacancies, stabilising the channel current to its initial state. Figure 3c shows

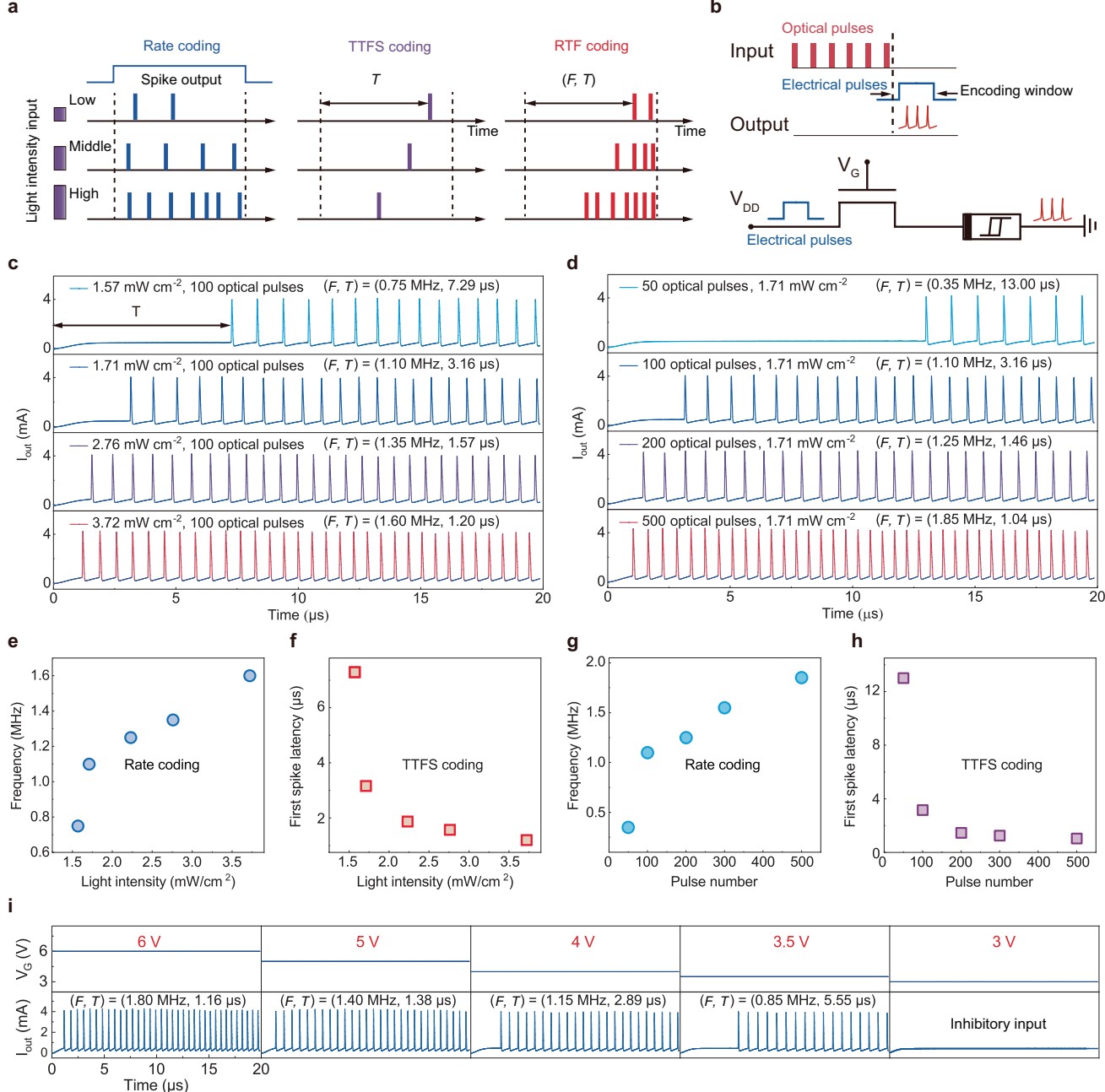

**Fig. 3 | Demonstration of rate-temporal fusion photoencoding. a** Schematic representations of biological neural coding models, including rate coding (blue line), TTFS coding (purple line), and RTF coding (red line). **b** Schematic circuit of 1T1R synapses and neurons. The optical pulse and the current waveform were regarded as the input and output signals, respectively. An electric pulse $V_{DD}$ (3 V, 20 μs) was applied to read the spike behaviour with a $V_G$ bias of 5 V. **c, d** Spiking behaviours of the 1T1R device triggered by various UV (**c**) illumination intensities and (**d**) optical pulse numbers. **e, f** The effect of the light intensity on the (**e**) spike frequency and (**f**) first spike latency. **g, h** The effect of the optical pulse number on the (**g**) spike frequency and (**h**) first spike latency. **i** Inhibitory voltage input can modulate the firing rate and the first-spike times.

the spiking behaviour of the artificial visual neuron in response to different light intensities with a fixed pulse number of 100. The spike frequency increased monotonically from 0.75 to 1.6 MHz as the light intensity increased from 1.57 to 3.72 mW/cm² (Fig. 3e). This behaviour can be attributed to the light-generated photocurrent, which leads to a decrease in $R_{ch}$. Importantly, as the light intensity increased, the first spike latency decreased from 7.29 to 1.2 μs (Fig. 3f). In addition, spiking behaviours related to the pulse number were demonstrated. Figure 3d shows the spike patterns obtained with various optical pulse numbers and a fixed light intensity of 1.71 mW/cm². Similarly, increasing the optical pulse number leads to an increase in the number of

photogenerated carriers due to the optoelectronic synaptic characteristics of the $In_2O_3$ phototransistor, resulting in lower $R_{ch}$. As a result, the artificial neuron has an increased spiking frequency (0.35–1.85 MHz) and a reduced first spike latency (13–1.04 μs) with increasing light pulse number (50–500), as shown in Fig. 3g, h, respectively. Thus, the visual stimuli information was successfully encoded into fast and precise electrical spikes via the artificial spiking visual neuron. This resembles the behaviour of biological visual neurons and demonstrates the potential of SNN with integrated rate and TTFS coding. Meanwhile, the $NbO_x$ Mott memristor with a smaller footprint (1 μm × 1 μm) and $In_2O_3$ phototransistor could be

monolithically integrated on the same chip with a conventional microfabrication process (Supplementary Fig. 12). The scale-down of the $NbO_x$ Mott memristor could further reduce both the light-induced firing threshold and power consumption. As a result, the RTF photo-encoding can be achieved under white light illumination without any initial light exposure (Supplementary Fig. 13).

In addition, the gate-controlled electrical properties of the synaptic phototransistor enable the representation of input stimuli with electrically configurable synaptic potentiation and depression behaviours; thus, the proposed device can mimic the process by which biological neurons recognise excitatory and inhibitory stimuli[43]. As shown in Fig. 3i, the spiking rate and latency can both be adjusted based on $V_G$. The spiking rate increased from 0.85 to 1.8 MHz, and the latency decreased from 5.55 to 1.16 µs (Supplementary Fig. 14) as $V_G$ increased (from 3.5 to 6.0 V) because of the increase in the channel conductance of the phototransistor (Supplementary Fig. 15). Thus, with its multiterminal configurability, the proposed artificial visual neuron could emulate the heterosynaptic plasticity of biological sensory neurons[42].

## Correlated synaptic plasticity and neural spiking dynamics

In biological sensory neural systems, stimulus information is represented via distributed neurons and synapse networks[43]. Synaptic plasticity describes the strength of communication between pre- and postsynaptic neurons in response to action potentials and is important in learning, memory, and forgetting[44]. In the human visual system (Fig. 4a), a visual neuron does not fire spikes until sufficient optical stimuli are received because the accumulated charge leaks away. When the accumulated charges (synaptic weights) contributed by the synapse surpass the threshold of the neuron, the neuron fires stimuli-relevant spike trains, enabling complex sensory and cognitive functions. Biological visual sensitisation occurs when repetitive series of brief stimuli are delivered at constant intensity, which increases perception sensitivity, facilitating precise sensory encoding and high responsiveness during dynamic visual processing[45–47].

To emulate biological synaptic plasticity, we utilise the artificial spiking visual neuron to achieve highly correlated synaptic plasticity and implement spiking neural dynamics at the device level. Based on optoelectronic synaptic plasticity (Supplementary Fig. 16), the $In_2O_3$ phototransistor can feasibly emulate the synaptic behaviours related to sensing, memory, and forgetting. Figure 4b illustrates the optical and electrical input schemes to reveal the correlations between optoelectronic synaptic plasticity and relevant neural spiking patterns. The optical input pulses were applied for a series of time windows ($t_0$–$t_4$, learning) and ceased, and the device was kept in the dark in the remaining time windows ($t_4$–$t_7$, forgetting). Moreover, electrical input pulses were applied throughout the process to record the output spike patterns.

The measured results are shown in Fig. 4c. Initially, the device is at rest and does not fire spikes without optical input ($t_0 = 0$ s). When a series of optical pulses is applied ($t_1 = 10$ s), obvious spiking patterns can be observed. The LIF neuron fires spikes when the voltage applied to the memristor reaches a threshold, showing activity-dependent RTF coding. As the applied time of the input optical pulses increased from 10 to 60 s ($t_1$–$t_3$), the spiking frequency increased from 1 to 1.55 MHz (Fig. 4e), and the first spike latency decayed exponentially from 3.51 to 1.19 µs (Fig. 4f). After the light pulse was stopped ($t_3 = 60$ s), the neuron continued spiking due to the LTP properties of the synaptic phototransistor, with a linearly decreasing spike frequency (1.55 to 0.80 MHz) and an exponentially increasing first spike latency (1.19–4.90 µs) over time ($t_3$–$t_6$, Fig. 4d). When the idle time is sufficient ($t_7 = 25$ min), the artificial neuron returns to the resting state and stops spiking. The spiking frequency and latency values after light illumination are shown in Fig. 4e, f, respectively. In addition, the 'forgetting' behaviours can be adjusted based on the input light intensity. In the

idle time, stronger input light intensity leads to neural spiking with both a longer period and longer latency, as shown in Fig. 4g, h, respectively. Furthermore, the RTF coding properties are related to the number of input light pulses (Supplementary Fig. 17). These correlated synaptic and neural spiking behaviours resemble natural learning and forgetting behaviours in a biologically plausible way at the device level.

To visualise the correlations between synaptic plasticity and the coding scheme, we constructed a simulation mushroom image map with $50 \times 50$ pixels to demonstrate the rate and TTFS coding results. The simulated mushroom image shows patterns of light illumination with intensities ranging from 1.71 to 3.73 mW/cm$^2$ (Fig. 4i). The rate and TTFS encoding data from Fig. 4g, h were utilised to create the map; the light intensity was encoded as the spike frequency within the range of 0–1.6 MHz, and the first spike latency was within the range of 0–10 µs. By incorporating synaptic plasticity, artificial visual neurons can achieve dynamic encoding across a broad range of time scales. With the rate coding scheme, because the spiking frequency related to the opto-synaptic weights decayed linearly (Fig. 4g), the image intensity gradually weakened over time without obvious contrast change (Fig. 4i, upper panel). In contrast, with the TTFS coding scheme, as the first spike latency increased exponentially, the image exhibited sharpened contrast as the intensity weakened over time (Fig. 4i, lower panel). Thus, by integrating both rate and temporal TTFS coding, the artificial visual neuron can biomimetically mimic human learning, memory, and forgetting behaviours.

## Implementation of rate and TTFS coding in SNN

We considered an autonomous driving task to demonstrate the advantages of our artificial visual neuron, including its sensory encoding and processing capabilities. Prediction tasks in complex traffic conditions, such as turning and overtaking at high speeds, require fast scene encoding and signal processing abilities[2]. An SNN with RTF coding and LIF neuron characteristics is proposed to satisfy these requirements. The RTF coding scheme has a high temporal resolution that is two orders of magnitude better than that of conventional image sensors. In addition, LIF neurons with temporal feature extraction abilities allow the SNN to process the input timing sequence signals better than other deep neural networks.

As shown in Fig. 5a, we utilised different coding schemes, including rate, TTFS, and RTF coding schemes, to encode the external road conditions as spike trains to implement the SNN. In the rate coding scheme, the light intensity of each pixel in an image is encoded as a spike train, where the spike number has a linear relationship with the light intensity (Supplementary Fig. 18). In the TTFS coding scheme, the input light intensity is converted into the first spike latency time, which follows an exponential decay law (Supplementary Fig. 19). In TTFS coding, the higher the value of the input intensity is, the earlier the spike firing time. In the RTF coding scheme, pixel values are encoded as spike trains with linear frequency and nonlinear temporal information (Supplementary Fig. 20). These three coding schemes are detailed in Supplementary Note 1–3. The multiplexed coding method has superior spiking temporal resolution and combines the advantages of the rate and TTFS coding schemes. Thus, the multiplexed coding scheme enables rapid and precise perception of visual stimuli in real-world environments.

As a demonstration, an SNN model based on ResNet-18 with LIF neurons is proposed to predict the steering angle and speed of an autonomous car[27,48–53]. The proposed SNN model is composed of 17 spike layers and one fully connected layer. The spike layer employs convolution kernels to compute membrane potential updates with the LIF neurons. These updates are subsequently integrated into the neurons' membrane potential and compared to a threshold to determine the spike firing rate ("Methods"). The external scene information, determined based on the public dataset (DAVIS Driving Dataset 2017),

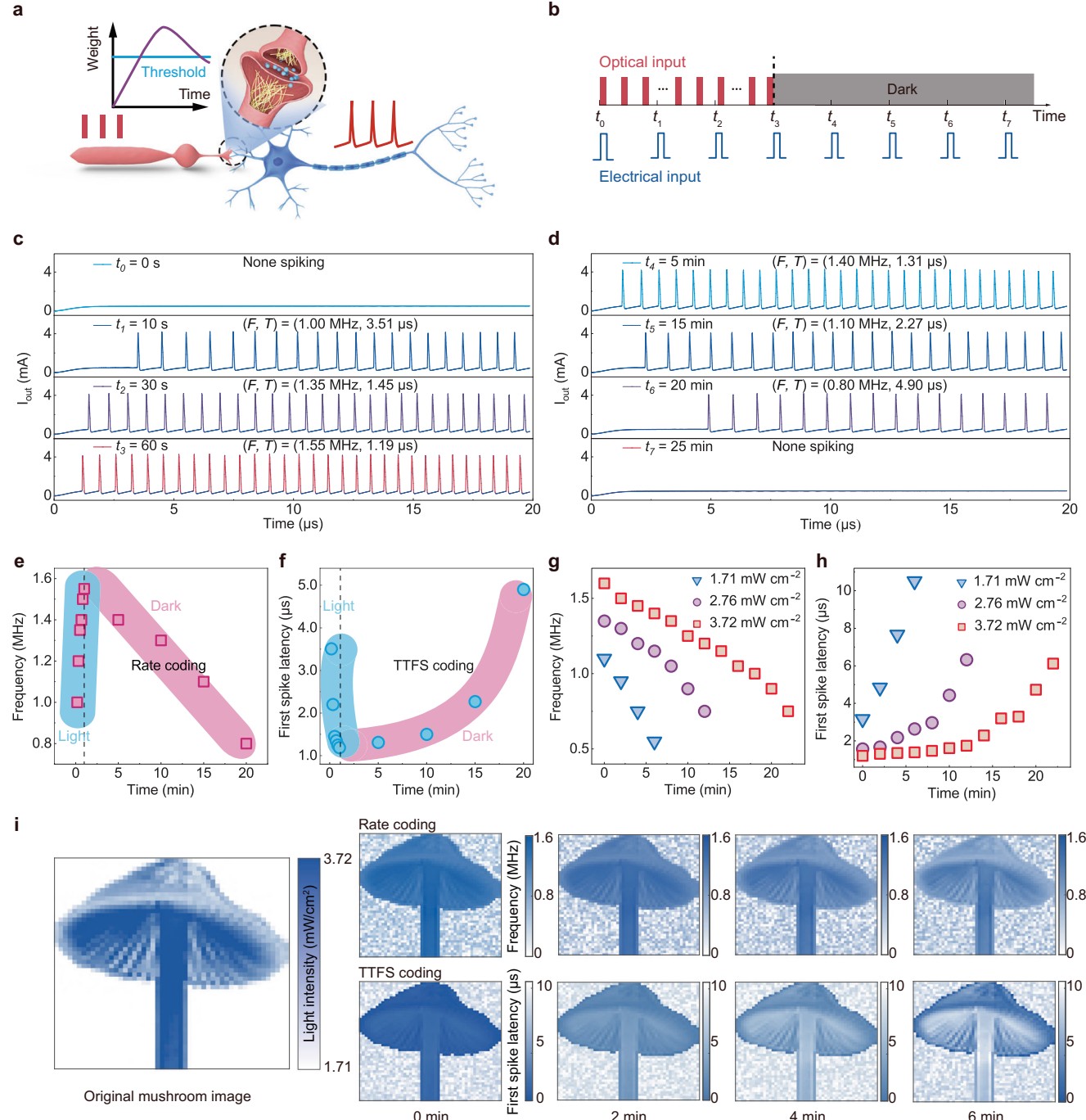

**Fig. 4 | Rate-temporal fusion scheme encodes synaptic plasticity in real-time for sensing and memory. a** Simplified schematic of sensitisation in the retina. **b** Operation scheme of optical (red line) and electrical input (blue line). UV illumination (pulse width: 5 ms, frequency: 5 Hz, intensity: 1.71 mW/cm² at 365 nm). Electrical pulses $V_{DD}$ (3 V, 20 μs) are applied to evaluate the spiking behaviour at $t_1$, $t_2$, $t_3$, and $t_4$, corresponding to 0, 10, 30, and 60 s, and $t_5$, $t_6$, $t_7$, and $t_8$, corresponding to 5, 10, 15, and 25 min. The gate voltage bias $V_G$ is 5 V. **c** Measured output current waveforms of sensitisation ($t_1$–$t_4$) with increasing synaptic weights under UV light illumination. **d** Measured output current waveforms of memory ($t_5$–$t_8$) with decreasing synaptic weights after the light pulse. **e**, **f** The sensing and memory processes have a linear relationship with the (**e**) spike frequency and an exponential relationship with the (**f**) first spike latency. **g**, **h** The RTF coding scheme exhibits changes in (**g**) spike frequency and (**h**) first spike latency during the memory window after different light intensities. **i** Image memory with rate coding and TTFS coding for a mushroom pattern at 0, 2, 4, and 6 min after light stimuli ceased.

comprises records of driving variables collected under different road conditions over 1000 km (Fig. 5a)[54]. This data is captured at a resolution of 100 × 140 pixels and encoded by our artificial visual neuron as 14k spike trains, which serve as inputs to the SNN model. The final output layer includes a single neuron, representing either the steering angle or the speed in various prediction tasks.

Two different road conditions, a turning road and a high-speed driving road, are selected to demonstrate the performance of the hardware-based SNN. As shown in Fig. 5b, the steering angle predicted by the RTF coding scheme is approximately equal to the ground truth. The rate coding scheme predicts a value similar to the ground truth, while the TTFS coding method cannot fit the ground truth. The loss

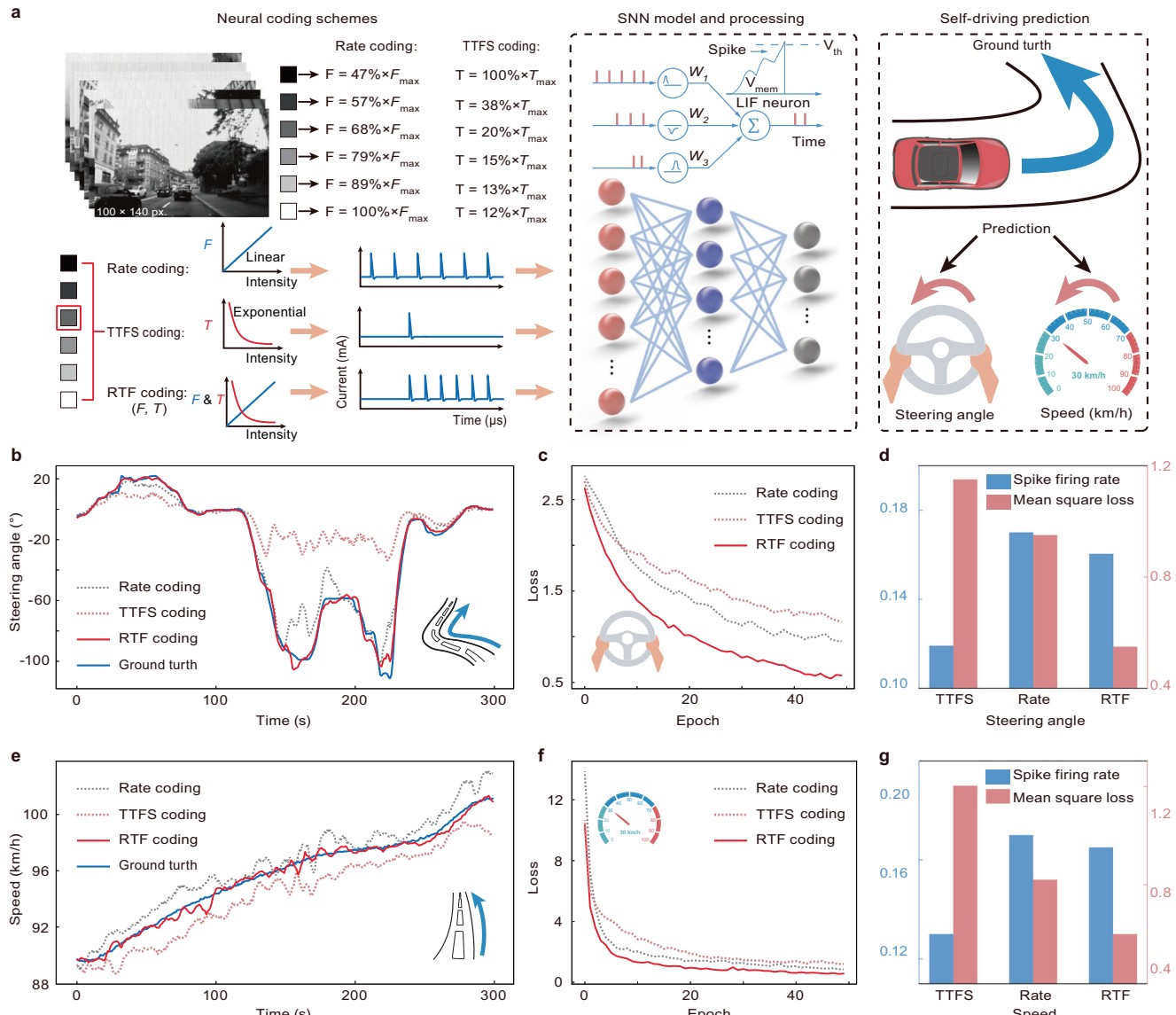

**Fig. 5 | Demonstration of the fused rate and TTFS coding scheme in a SNN to predict the speed and steering angle of a self-driving vehicle. a** The intensity of the pixels is encoded in terms of the firing frequency and first spike latency of the artificial visual neuron. Greyscale images were encoded as several series of spike trains by the rate coding, TTFS coding, and RTF coding schemes, and fed into a trained SNN to predict the vehicle speed and steering angle. **b** Comparison of the steering angle prediction results by the rate coding, TTFS coding, and RTF coding schemes in a scenario with complex road corners. **c** Loss value per epoch during training of the SNN for steering angle prediction under different coding schemes. **d** Comparison of three coding schemes in terms of the spike firing rate and loss in predicting the steering angle. **e** Comparison of the speed prediction results of the rate coding, TTFS coding, and RTF coding schemes in a scenario with smooth road corners. **f** Loss value per epoch during training of the SNN for speed prediction under different coding schemes. **g** Comparison of the spike firing rate and loss in the speed prediction task.

curves of different coding techniques for predicting the steering angle are depicted in Fig. 5c. The final loss of the rate coding scheme is lower than that of the TTFS coding scheme because the TTFS coding scheme emits only one spike per pixel and thus cannot provide sufficient information for the processing algorithm. In comparison, the RTF coding scheme with linear frequency and nonlinear temporal characteristics achieves the best performance, with a loss of 0.5 after 50 epochs.

As shown in Fig. 5d, TTFS coding with a lower spike firing rate leads to a higher mean square loss, which represents the poor fitting performance between the predicted results and the recorded values in the dataset. The rate coding scheme with a multispike encoding mechanism leads to a lower loss between the predicted values and ground truth. The RTF coding scheme with a spike firing rate similar to the rate coding scheme leads to a significantly decreased loss, achieving the best fitting performance for the recorded vehicle parameters of the driving car. In addition, as the change in the speed of the

self-driving car is small at the curves in the road, the loss functions differ only slightly among the three encoding methods (Supplementary Fig. 21).

As shown in Fig. 5e, the comparison results indicate that the speed values predicted by the RTF coding scheme are similar to the ground truth when driving at high speeds. The RTF coding scheme has a lower loss than the TTFS and rate coding schemes, which proves the superiority of this fusion coding method with frequency and temporal characteristics (Fig. 5f, g). Similarly, high-speed driving roads have smaller corners, and the loss values of the steering angle predictions with the three coding schemes are approximately the same (Supplementary Fig. 22). Overall, based on the rapid and precise prediction ability of the RTF coding scheme and the temporal feature extraction ability of LIF neurons, our proposed SNN could accurately predict the steering angle and speed of autonomous vehicles in various tasks under different road conditions.

## Discussion

In summary, an artificial visual spiking neuron composed of an $In_2O_3$ synaptic phototransistor and a $NbO_x$ memristor-based LIF neuron was experimentally demonstrated. The artificial neuron enables multiplexed rate and TTFS coding of external visual information. The proposed RTF encoding scheme can achieve precise timing and rapidly and accurately encode the original input information. The artificial neuron has fast spike latency coding from 13.00 to 1.04 μs, tunable firing frequency coding from 0.35 to 1.85 MHz, low energy consumption (1.06 nJ/spike), high endurance (>$10^{10}$), and multiplexed information encoding capabilities. The RTF encoding scheme results are consistent with real-world ground truth data, and an SNN with the proposed RTF coding scheme achieves high accuracy for steering and speed prediction for self-driving vehicles. The complementary coding capability of the artificial neuron ensures rapid and precise perception capability in complex environments, demonstrating the high efficacy of SNN.

## Methods

### Preparation of $NbO_x$ memristors and $In_2O_3$ optoelectronic synaptic transistors

The $NbO_x$ memristors were fabricated as follows: After photolithography and lift-off processes were applied, the bottom Ti (5 nm)/Pt (35 nm) electrodes were deposited on a Si/SiO$_2$ substrate by e-beam evaporation. Then, 35 nm $NbO_x$ active layers were deposited by magnetron sputtering, and patterned with photolithography and lift-off processes at room temperature. Afterwards, the top Ti (5 nm)/Pt (35 nm) electrodes were grown by e-beam evaporation and patterned by photolithography and lift-off processes. The two-terminal metal-insulator-metal structure was integrated into the source of the transistor. The Mott memristors have a working area of 7 μm × 7 μm.

The $In_2O_3$ optoelectronic synaptic transistors were fabricated as follows: A precursor solution of $In_2O_3$ with a concentration of 0.1 M was prepared by dissolving indium nitrate in 2-methoxyethanol (2-ME). To enhance the exothermic combustion reaction, acetylacetone (AcAc) and ammonium hydroxide ($NH_3 \cdot H_2O$) were added to the solution in equimolar quantities to indium nitrate. The solution was vigorously stirred overnight and filtered using a 0.2 μm syringe filter before utilisation. A substrate consisting of a highly doped (p$^{++}$) silicon wafer with a 100 nm thermally grown $SiO_2$ layer was employed. Following a UV-ozone treatment for 10 min, the precursor solution was spin-coated at a speed of 3000 rpm for 45 s, and subsequently, the device was prebaked at 100 °C. The device was then baked at 200 °C for 5 min, and conventional photolithography was performed. The pattern was achieved by etching in a mixed solution of diluted hydrochloric acid and deionized water (1:15, v-v) for 5 s. The $In_2O_3$ channels were then annealed in air at a temperature of 300 °C for one hour. Finally, the Ni/Au (8 nm/50 nm) source/drain electrodes were thermally evaporated utilising the lift-off process to obtain a channel width/length (W/L) of 400 μm/10 μm.

### Device characterisation

The cross-section TEM and HRTEM images were obtained with a transmission electron microscope (FEI Tecnai TF-20, UK) and analysed. Room-temperature electrical measurements were carried out using a Summit 1100B-M Probe Station. The DC mode was measured by an Agilent B1500 semiconductor parameter analyser. A B1530A fast measurement unit module was used to simultaneously generate the voltage pulse and measure the response current.

### SNN processing framework for the autonomous driving task

The processing framework consists of neural coding techniques and the SNN model. Three different coding techniques, including rate, TTFS, and RTF coding, were adopted in this task. The frame signal of the road conditions was obtained from a public dataset (DAVIS Driving Dataset 2017). The rate coding scheme converts each pixel value in the frame into a spike train with a firing rate proportional to the light intensity. In addition, the generated spikes follow a Poisson distribution. The TTFS coding scheme utilises only one spike to encode the light intensity of each pixel, and the firing time attenuates exponentially with increasing light intensity. With this scheme, higher light intensity leads to earlier spike firing times. The RTF scheme coding combines the characteristics of the above two methods, and the input pixel grey values are encoded as pulse trains with a firing time (1.2–20.0 μs) and firing rate (0–1.6 MHz). Finally, these three coding schemes are all implemented over 15 time steps.

The SNN model used in this task has a ResNet-18 architecture, and LIF neurons are used to process the input spikes. The numbers of LIF neurons in layers 1, 2–5, 6–9, 10–13, and 14–17 are 14 k, 896 k, 448 k, 160 k, and 120 k, respectively. LIF neurons have dynamic features and are thus powerful and energy-efficient in predicting the steering angle and speed. The updated value of the membrane potential is calculated by inputting the presynaptic neuronal spikes $Xi$ of the neuron multiplied by the synaptic weights $Wi$, and the membrane potential of the postsynaptic neuron in the next layer is calculated as follows:

$$V_{mem}(t+1) = \beta V_{mem}(t) + \sum_{i=1}^{k} W_i^T X_i(t) - R\left[\beta V_{mem}(t) + \sum_{i=1}^{k} W_i^T X_i(t)\right]$$
(1)

$$R = \begin{cases} 1, & if\ V_{mem} > V_{th} \\ 0, & otherwise \end{cases}$$
(2)

where $\beta$ and $k$ are the membrane potential decay rate and the number of neurons in this layer, respectively. $T$ is the transposition operation. If the membrane potential of the LIF neuron is more than the threshold, the neuron generates a spike, which is used as the input to the next layer. Then, the membrane potential is reset to zero.

In this experiment, the mean absolute error (MAE) loss function is adopted to evaluate the difference between the predicted value and the ground truth label in the dataset. The loss is computed as follows:

$$Loss(x, y) = \frac{1}{n}\sum_{i=1}^{n}\left|y_i - x_i\right|$$
(3)

Then, to train the proposed SNN, a surrogate gradient descent algorithm−backpropagation through time (BPTT)−is used to update the synaptic weights. In the BPTT scheme, the network is unrolled in the time dimension to calculate the weight update value. The details of the weight update process are as follows:

$$\Delta w^l = \sum_n \frac{\partial L_{total}}{\partial o_t^l} \frac{\partial o_t^l}{\partial V_t^l} \frac{\partial V_t^l}{\partial w^l}$$
(4)

$$\frac{\partial o_t^l}{\partial V_t^l} = \begin{cases} H_1'(V_i - V_{th}), & if\ o_t^l = S_t^l \\ 1, & if\ o_t^l = V_t^l \end{cases}$$
(5)

where $w^l$ is the weight in the spike layers, and $L_{total}$ is the total loss between the prediction and the ground truth label in the dataset. $o_t^l$, $V_t^l$ and $w^l$ are the output of the neuron, the membrane potential of the LIF neuron at time $t$, and the weight in the spike layers. $S_t^l$ is the output spike of the LIF neuron at time $t$. The shifted Arctan function $H_1(x) = \frac{1}{\pi}\arctan(\pi x) + \frac{1}{2}$ is used as the surrogate function to replace the Heaviside function of the LIF neuron during the backpropagation process. In addition, the dataset consists of 20 k scene frames, which are divided into training and testing sets. The first 80% of the data are used to train the model, and the remaining 20% of the data are used to validate the model's performance. Finally, we use dropout and regularisation to prevent overfitting and improve model performance.

## Data availability

The data that support the plots within this paper and other findings of this study are available from the corresponding author upon request. Source data are provided with this paper.

## Code availability

The code for the SNN with the proposed RTF coding scheme is available from the corresponding author with detailed explanations upon request.

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

## Acknowledgements

This work was supported by the National Natural Science Foundation of China (Grant No. 62174138) and the Key Project of Westlake Institute for Optoelectronics (Grant No. 2023GD004). We thank the Westlake Centre for Micro/Nano Fabrication, the Instrumentation and Service Centre for Physical Sciences (ISCPS), and the Instrumentation and Service Centre for Molecular Sciences (ISCMS) at Westlake University for facility support and technical assistance.

## Author contributions

B.Z., H.W., and F.L. conceived and designed the experiments. F.L. and D.L. designed and fabricated the $In_2O_3$ phototransistor and $NbO_x$ memristor. C.W. and R.W. conducted the simulation. G.L. designed and conducted the XPS measurements and analysis. F.L., D.L., G.L., H.R., Y.T., Y.W., Y.C., and K.L. carried out the electrical experiments and characterised the devices. F.L, Q.H., M.S., M.Q., H.W., and B.Z. analysed the data and discussed the results. All authors contributed to the preparation of the manuscript. B.Z. and H.W. supervised the project.

## Competing interests

The authors declare no competing interests.
