## [Peer Review File · Nature Communications]

Reviewers' comments:

Reviewer #1 (Remarks to the Author):

The author proposes an artificial visual neuron with multiplexing rate and first peak encoding time. Although SNN related work may attract people's interest, the novelty of this manuscript can not reach the high criteria of Nature Communications. Here are some specific comments on the manuscript:

1. Recently, many studies have proposed the use of memristors as synaptic devices, and the manuscript used the device configuration integrating with In₂O₃ optoelectronic synaptic transistors and NbOx Mott memristors is common.

2. The encoding schemes mentioned in the article are two commonly used methods in SNN. For the implementation of RTF encoding function, only two different encoding methods were integrated, without demonstrating unique innovation, and in the entire work, the participation of the equipment was not demonstrated, which cannot reflect the necessity of implementing RTF encoding by the equipment. In addition, the SNN related results in the manuscript were obtained through computer simulation and were not implemented in hardware.

3. There are some unclear descriptions in the manuscript, such as how to initialize the state of a phototransistor? What are the specific values of the missing 'R_m' and 'R_i' parameters; In addition, the specific curve of the change on "R_{ch}" is not shown.

4. In Figure 1 (b), in Time-to-first-spike coding, the first peak arrival time obtained from high light intensity stimulation is longer than that obtained from low light intensity stimulation, which is inconsistent with the description. What is the reason for this?

In summary, the advance of this work is too small to be published in Nature Communications, and it is recommended to transfer to other journals.

Reviewer #2 (Remarks to the Author):

The authors have developed an artificial spiking neuron that integrates an In₂O₃ synaptic phototransistor and an NbOx Mott memristor to mimic biological photoreceptors and retinal ganglion neurons for visual information processing. The key accomplishment, in my view, is the ability to achieve rate and TTFS coding without the need for auxiliary reset circuits. The authors also employed rate and temporal fusion (RTF) coding. While the study is interesting, I have several comments that must be addressed before a decision on publication can be made for high-impact journals like Nature Communications

While I do understand that the oscillation frequency of the Mott insulator is determined by the R_{ch} of the In₂O₃ phototransistor, I do not understand why the time to oscillation differs. The authors are requested to provide a clear explanation. It would be nice if the authors used some semi-empirical or physics-based models to explain the device characteristics.

The authors used an optical laser to provide light stimuli of a fixed wavelength of 365 nm (UV). Whereas their practical application involves ambient lighting comprising various visible wavelengths. Therefore it is important to demonstrate the spectral response of the proposed device. Also, I would recommend repeating the experiments with LEDs or more realistic diffused light sources than monochromatic and

focused UV light beams.

What illumination biasing conditions were used for the In₂O₃ phototransistor? Different biasing conditions for write and read operation will necessitate the use of peripheral circuits. Authors must comment on and address this issue.

It is not clear how the authors obtained data for 10¹⁰ cycles in Fig. 2c. How long each sweep took? My biggest concern is reproducibility. What is the device-to-device variation in NbOx Mott memristor and In₂O₃ phototransistor?

It is also not clear whether both devices are fabricated on the same chip or externally connected. I will be more excited if the two devices are integrated on the same chip.

Which tool was used to measure the switching characteristics of NbOx Mott memristors? As far as I know, most advanced parameter analyzers have resolutions of 10s of nanoseconds. How did the authors make such precise measurements?

Responses to NCOMMS-23-46955

REVIEWER COMMENTS

Reviewer 1:

1. Recently, many studies have proposed the use of memristors as synaptic devices, and the manuscript used the device configuration integrating with In_2O_3 optoelectronic synaptic transistors and NbO_x Mott memristors is common.

Response: Thank you for your comment.

Memristors could be classified into two categories: non-volatile and volatile memristors. On the one hand, the **non-volatile memristors** have been widely used as memory and **synaptic devices**, by virtue of their configurable non-volatile characteristics. Many efforts were dedicated to emulating essential synaptic functions within the visual neural system and developing light-sensitive synaptic devices aimed at mimicking short-term/long-term memory related to light.

On the other hand, artificial spiking neurons have garnered widespread interest, given the general consensus that information inherent in event-driven rate coding is highly energy-efficient and robust against noise¹⁻⁶. In this regard, **volatile memristors such as NbO_x** , with typical threshold switching behaviors, could be implemented as **artificial neuron devices**, performing more complicated neural functions such as integration and firing spikes.

Although various synaptic devices have been reported, high-performance neural devices with smaller footprint, high biological plausibility, and multiplexed coding schemes, remain limited.⁷ In this work, we constructed an artificial visual spiking neuron by implementing **NbO_x Mott memristors as artificial neuron devices, not as synaptic devices**. The artificial visual spiking neuron exhibits light-intensity-dependent spiking frequency output and an ultra-high time resolution of approximately 1 μs , allowing for the rapid and precise encoding of temporal information from visual stimuli. More importantly, we achieved both rate coding and time-to-first-spike (TTFS) coding, for the first time, at the hardware level in our visual spiking neuron, signifying the efficacy of spike-based neuromorphic hardware.

In comparison, in recent studies on neuromorphic sensory perception and computation, complex transistor-based analog-to-digital conversion (ADC) circuits are required to convert synaptic post-synaptic currents into digital signals, such as in “*Mammalian-brain-inspired neuromorphic motion-cognition nerve achieves crossmodal perceptual enhancement. Nat. Commun.* 14, 1344, (2023)”. However, these CMOS-based neuron implementations face challenges related to scalability (capacitors are required), limited neuronal dynamics, and suboptimal energy efficiency when compared to biological neural systems.

Our artificial visual neurons offer high scalability and a compact structure. The synaptic transistors within our architecture enable in-sensor computation, simulating the sensitization and memory functions of biological synapses. The neuronal devices feature threshold-controlled digital sensing signal output, multiplexed rate and time-to-first-spike coding functions. In contrast to hardware that relies on complex transistor circuitry to emulate bio-inspired functions, our single devices, with such intricate behavior, can replace hundreds or even thousands of transistors in terms of functionality, offering higher-level features and energy efficiency through their inter-device interactions.

Still, to highlight the advances made in this work, a benchmarking table on artificial spiking visual neurons is shown in Supplementary Table 2. Our device shows clear advantages in neuron coding capabilities, spiking frequency, and time resolution, etc.

Supplementary Table 2. Comparison of characteristics of various artificial spiking visual neurons.

Rate	Neuron coding		Excitatory and inhibitory	Spike integration	Synaptic plasticity	Spiking frequency (Hz)	Spiking time resolution (s)	Energy/spike	Ref.
	TFS	TRF							
Yes	No	No	No	No	No	0-10 ⁵	No	NA	⁸
Yes	No	No	No	Yes	No	0.1-1200	No	~40 pJ	⁵
No	Yes	No	Yes	Yes	Yes	9.6	~0.23	~93 mJ	⁹
Yes	No	No	Yes	Yes	No	35-60	No	~6.8 nJ	¹⁰

Yes	No	No	No	Yes	No	1-200	No	~2.5 nJ	¹¹
Yes	No	No	No	Yes	Yes	0-160	No	NA	¹²
No	Yes	No	No	Yes	No	No	0.7	0.1-1 μ J	⁶
Yes	Yes	Yes	Yes	Yes	Yes	(0.35-1.85) $\times 10^6$	1.04×10^{-6}	1.06 nJ	This work

2. The encoding schemes mentioned in the article are two commonly used methods in SNN. For the implementation of RTF encoding function, only two different encoding methods were integrated, without demonstrating unique innovation, and in the entire work, the participation of the equipment was not demonstrated, which cannot reflect the necessity of implementing RTF encoding by the equipment. In addition, the SNN related results in the manuscript were obtained through computer simulation and were not implemented in hardware.

Response: Thank you for your feedback. We agree that the two encoding schemes mentioned in the article are commonly used in SNN. However, we achieved them at the device level by a single artificial neuron device, rather than CMOS circuits. Such a hardware implementation with emerging devices was challenging and had not been accomplished before. Our work provides unique approaches to emulate the multiplexed encoding of biological neural systems with simple electronic devices, offering alternative and more biologically plausible devices beyond conventional CMOS electronics.

In our work, we elucidate the significance of implementing rate and temporal fusion (RTF) encoding scheme, particularly in the context of SNN. In general, rate coding is usually more robust than TTFS coding. Rate coding represents information through the spike frequency generated by neurons, making it relatively insensitive to minor variations in input. Therefore, it performs more robustly in the presence of noise or significant environmental changes. However, TTFS coding relies on the first spike time of the neuron upon stimulus onset, making it more sensitive to minor variations in input. In urgent situations, TTFS coding is superior to rate coding in reliability because rate coding cannot provide efficient temporal information or sufficient features to fully represent the stimulus, especially in extremely short time scales. In complex

environments, often characterized by noise, unpredictable changes, and transient scenes, ensuring efficient perception is crucial. Therefore, the utilization of multiplexed rate and TTFS coding becomes essential.

In the realm of bio-inspired or neuromorphic computing devices, hardware approaches often rely on complex transistor circuits to simulate biological functionalities¹³. For example, “*A Sparse and Spike-Timing-Based Adaptive Photoencoder for Augmenting Machine Vision for Spiking Neural Networks. Adv. Mater.* 34, e2202535, (2023)” employs a circuit consisting of 21 transistors to emulate time-to-first-spike (TTFS) encoding behavior. The implementation of neuronal functionalities in SNN, such as rate and TTFS coding, heavily depends on software or digital circuitry, with different encoding methods necessitating various algorithmic and circuitry modules. Therefore, software-based neurons require substantial computational resources and power budgets.

Therefore, to unlock the full potential of advanced computing architectures such as SNN, there is an urgent demand for artificial neuronal devices featuring a range of dynamic encoding functionalities to optimize information processing in neuromorphic hardware. The RTF coding scheme can compensate for the temporal information missing in rate encoding, offering increased robustness to noise and superior efficiency in spike counting. The accuracy of SNNs in tasks like self-driving vehicles prediction significantly depends on the RTF encoding precision in device spike sequences, which, in turn, affects their ability to process visual data rapidly and accurately in complex visual environments.

In the short term, we may not have the capabilities and/or equipment to implement such complex autonomous driving tasks with our device based SNN. However, our simulation data signifies the feasibility of our approach, and demonstrates that our complementary encoding method could ensure more rapid and accurate perception in complex environments. Furthermore, recent hardware-based neuron applications have also validated the potential of single-device simulations for their applications. Examples include “*Atomically thin optomemristive feedback neurons. Nat. Nanotechnol.* 18, 1036-1043, (2023)”, “*Selective area doping for Mott neuromorphic*

electronic. Sci. Adv. 9, eade4838, (2023)”, and “*A two-dimensional mid-infrared optoelectronic retina enabling simultaneous perception and encoding. Nat. Commun.* 14, 1938, (2023)”. These works showcase the potential and versatility of emerging materials in hardware-based neuromorphic electronics.

We appreciate your feedback and are continuously exploring avenues to translate these simulations into practical hardware implementations for more complex tasks.

3. There are some unclear descriptions in the manuscript, such as how to initialize the state of a phototransistor? What are the specific values of the missing 'R_m' and 'R_i' parameters; In addition, the specific curve of the change on "R_{ch}" is not shown.

Response: To initialize the state of a phototransistor, an electric pulse (V_G: 10 V, 20 μs) is applied to the gate electrode. This bias voltage helps to restore the ionized oxygen vacancies, stabilizing the channel current to its initial state. Typically, the recovery time required depends on the intensity of light or the number of light pulses, with higher light intensity or more light pulses requiring a longer recovery time.

The specific values of the missing “R_m” and “R_i” parameters are as follows:

$$R_i = \frac{V_{th}}{I_{th}} = 5349.5 \Omega \quad (1)$$

$$R_m = \frac{V_{hold}}{I_{hold}} = 293.3 \Omega \quad (2)$$

The R_i and R_m are the insulating and metallic resistances of the NbO_x memristor, corresponding to the resistance values of high resistance state (HRS) and low resistance state (LRS), respectively. To avoid confusion, we now use “R_{HRS}” and “R_{LRS}” instead of “R_m” and “R_i”.

As shown in Figure R1, following your suggestion, we extracted the specific curve of the change in “R_{ch}” with:

$$R_{ch} = \frac{V_{DS}}{I_{DS}} \quad (3)$$

Figure R1. The change in R_{ch} of In_2O_3 phototransistor as a function of light power density (from 1.57 to 4.37 mW/cm^2 , $\lambda = 365 \text{ nm}$). R_{ch} measured at a DC bias of $V_{DS} = 3 \text{ V}$ and $V_{GS} = 5 \text{ V}$.

In addition, we added the following descriptions to the revised manuscript (paragraph 1, page 6, and paragraph 2 and 4, page 8, highlighted): “After each test, an electrical pulse ($V_G: 10 \text{ V}$, $20 \mu\text{s}$) is applied to the gate electrode of the phototransistor to initialize its state. This process ensures the restoration of ionized oxygen vacancies, stabilizing the channel current to its initial state.” “In addition, R_{LRS} is 293.3Ω , and R_{HRS} is 5349.5Ω .” “When V_{DD} and V_G are fixed, R_{ch} is determined by the parameters of the optical pulses (Supplementary Fig. 10).”

4. In Figure 1 (b), in Time-to-first-spike coding, the first peak arrival time obtained from high light intensity stimulation is longer than that obtained from low light intensity stimulation, which is inconsistent with the description. What is the reason for this?

Response: Appreciate your careful inspection. We deeply apologize for this issue, which was due to an error in the illustration. After a more thorough review of the data and charts, we identified this mistake and made the necessary corrections. Now, Figure 1(b) accurately represents the difference in time-to-first-spike coding between high light intensity and low light intensity stimulation.

Reviewer 2:

1. While I do understand that the oscillation frequency of the Mott insulator is determined by the R_{ch} of the In_2O_3 phototransistor, I do not understand why the time to oscillation differs. The authors are requested to provide a clear explanation. It would be nice if the authors used some semi-empirical or physics-based models to explain the device characteristics.

Response: Thanks for your suggestion. We apologize for the confusion caused.

The R_{ch} of the In_2O_3 phototransistor could influence both the oscillation frequency and the time to first-spike: lower R_{ch} leads to higher frequency and shorter time to first-spike. To provide a clearer explanation, we analyzed and illustrated the relationship between the first-spike time and R_{ch} in Figure R2.

Figure R2. **a**, A small-signal model of an oxide TFT. **b**, The equivalent circuit of the TFT comprises a resistor and a capacitor connected in parallel, without considering contact resistance, and is then linked to a constant current source. **c**, The change in current across the equivalent circuit resistance. **d**, Schematic circuit of 1T1R synapses and neurons. **e**, The optical pulse and the current waveform were regarded as the input and output signals, respectively. An electric pulse (V_{DD} : 3 V, 20 μ s) along with a V_G electric pulse (3 V, 20 μ s) was applied to read the spike behaviors.

Figure R2a shows an equivalent model of thin film transistor (TFT), where C_{GS} and C_{GD} are the gate-source and gate-drain capacitance, respectively. g_m is the transconductance of TFT, and g_{ds} is the channel conductance. R_D and R_S are the drain and source contact resistance, respectively. The equivalent circuit of the TFT, without

considering contact resistance, is shown in Figure R2b. The parallel R-C circuit offers a better representation of the neuronal cell membrane. According to the RC charge and discharge circuit, we can get:

$$I_R(t) = I(1 - e^{-\frac{t}{RC}}) \quad (4)$$

Here, R and C are the equivalent channel resistance and capacitance of the series circuit respectively. As shown in Figure R2c, after the transistor is turned on, I gradually increase to its maximum value.

In Figure R2d, the source of the TFT and the memristor are connected in series. In this series circuit, as light pulse number increases (Figure R2e), R_{ch} decreases (365 nm, 1.71 mW/cm², 5 ms, 0, 20, 40, and 100 cycles correspond to R_{Dark} , R_{ch1} , R_{ch2} , and R_{ch3} , respectively), resulting in a smaller time constant ($\tau_{integration}$). This, in turn, leads to a faster current reaching the firing threshold, resulting in the quicker arrival of the first spike (shorter time to first-spike). As shown in Figure R2e, R_{ch3} has the lowest resistance value, and it leads to faster generation of spikes within the 5 μ s time window; in comparison, R_{dark} , R_{ch1} , and R_{ch2} , with larger resistance values, didn't generate spikes in the same time window. This proves that lower R_{ch} results in shorter time to first-spike.

We added the following statements to the revised manuscript: “*The spiking behaviours of the artificial neuron can be altered by adjusting the R_{ch} values based on a leaky integrate-and-fire (LIF) model, leading to different spiking durations $\tau_{integration}$ ($\sim R_{ch}C$) (Supplementary Fig. 11).*”

2. The authors used an optical laser to provide light stimuli of a fixed wavelength of 365 nm (UV). Whereas their practical application involves ambient lighting comprising various visible wavelengths. Therefore, it is important to demonstrate the spectral response of the proposed device. Also, I would recommend repeating the experiments with LEDs or more realistic diffused light sources than monochromatic and focused UV light beams.

Response: We thank the reviewer for this comment.

In Supplementary Fig. 8, we have already verified the absorption of visible light

by the In_2O_3 film. In response to your concerns, we conducted additional experiments to showcase the response of the In_2O_3 phototransistor to white light. We made new monolithically integrated devices with 1-memristor-1-phototransistor structures on the same chip and characterized their performance.

Figure R3a illustrates the transfer characteristics of the In_2O_3 phototransistor under different light intensities (Agilent B1500). Furthermore, we have taken your suggestion into account and repeated the experiments of rate-temporal fusion photo-encoding using white light LED instead of monochromatic and focused UV light beams (Figure R3b-d).

Figure R3. a, Transfer curves of the In_2O_3 phototransistor as a function of light power density (from 2.9 to 19.5 mW/cm², white light). I_{DS} versus V_{GS} measured at a drain bias of $V_{DS} = 3$ V. **b,** The change in R_{ch} of In_2O_3 phototransistor as a function of light power density ($V_{DS} = 3$ V and $V_{GS} = 3$ V). **c,d,** The effect of the light intensity on the (c) spike frequency and (d) first spike latency.

As you correctly point out, sensory information originates not only from the visible light spectrum but also from other spectral regions, including ultraviolet (UV). Several species of insects, like butterflies, fish, and birds, possess the ability to perceive UV light, granting them tetrachromatic vision. For instance, butterflies use UV light

patterns to distinguish stamen, pistil, and petals in flowers, allowing them to accurately locate their target. Exploring the perception of UV light by neuro-morphic vision sensors holds significant value, especially in UV radiation warning applications¹⁴. Therefore, we firmly believe that retina-inspired optoelectronic devices need not be confined solely to processing visible information for human visual perception but extending their operating range to the UV spectrum can offer substantial benefits in the context of advanced machine vision.

We added the following statements to the revised manuscript (paragraph 1, page 9): “*Additionally, rate-temporal fusion photo-encoding can be achieved under white light conditions, without any initial light exposure (Supplementary Fig. 13).*”

3. What illumination biasing conditions were used for the In₂O₃ phototransistor? Different biasing conditions for write and read operation will necessitate the use of peripheral circuits. Authors must comment on and address this issue.

Response: The In₂O₃ phototransistor does not require changing biasing conditions during illumination. To read the spike behavior, a particular biasing condition is employed for the gate electrode and the transistor's drain terminal. This involves the application of an electric pulse (V_{DD} : 3 V, 20 μ s) along with a V_G electric pulse (3 V, 20 μ s) to capture the current waveform.

4. It is not clear how the authors obtained data for 10¹⁰ cycles in Fig. 2c. How long each sweep took?

Response: The data for the 10¹⁰ cycles shown in Fig. 2c was obtained through a programmed pulsed test by a B1500 system, examining the endurance characteristics of the NbO_x Mott memristor (Figure R4).

The device was applied with periods of pulse trains with a pulse width of 1 μ s and period of 10 μ s until 10¹⁰ cycles. Following the completion of the expected number of pulse cycles (at 10ⁿ cycle (n = 0, 1, ..., 10)), we utilized pulsed read voltage program to capture the momentary current transition. This approach allowed us to precisely measure and record the behavior of the NbO_x Mott memristor at the end of the

endurance test.

Figure R4. Pulse operation with the endurance characteristics of the NbO_x Mott memristors.

We added the following statements to the revised manuscript: “*Pulse operation with the endurance characteristics of the NbO_x Mott memristors is further illustrated (Supplementary Fig. 5).*”

5. My biggest concern is reproducibility. What is the device-to-device variation in NbO_x Mott memristor and In₂O₃ phototransistor?

Response: Thanks for the insightful comment and we appreciate your concern regarding device-to-device variation. We have conducted more experiments to evaluate the device-to-device variation.

Figure R5 shows the characterization of 100 artificial neurons on a 2-inch silicon wafer. The variations in threshold switching characteristics and electroforming operation of 100 NbO_x Mott memristors with a working area of 1 μm × 1 μm are presented in Figure R5a and R5b, respectively. To investigate the uniformity among devices, statistical measurements and extractions were conducted for the voltage distribution (Figure R5c) and resistance distribution (Figure R5d) of the 100 NbO_x memristors. The calculated coefficient of variation (C_v) for different parameters (V_{th}, V_{hold}, and V_{forming}) is shown in Figures R5e-f, indicating device-to-device variabilities of 3.49%, 3.03%, and 2.33%, respectively. These measurements were conducted by an Agilent B1500 system.

These results suggest that NbO_x Mott memristors exhibit excellent uniformity, with device-to-device variation lower than that of previously published volatile threshold switching devices.

Meanwhile, Figure R5h illustrates the transfer characteristics of 100 In₂O₃ phototransistors with channel length of 10 μm and channel widths of 400 μm. Figure R5i-l present the corresponding device-to-device variation in μ_{sat}, V_{th}, SS, and I_{on/off}, with values of 0.19, 0.36, 0.21, and 0.52, respectively. In conclusion, the artificial neurons in this study demonstrate a low variability. The TFT measurements were conducted by a Keithley 4200A SCS system.

Figure R5. Device-to-device variation of 100 artificial visual neurons on 2-inch silicon wafer. **a**, Current-voltage (I-V) curves with threshold switching characteristics of 100 NbO_x Mott memristors with a working area of 1 μm × 1 μm. **b**, Electroforming operation of the 100 NbO_x memristor. **c**, The V_{th}, V_{hold} and V_{forming} of 100 NbO_x memristors. **d**, Device-to-device variation of R_{LRS} and R_{HRS} for 100 devices. **e-g**, The statistical analysis of variability of V_{th} (**e**), V_{hold} (**f**) and V_{forming} (**g**) in 100 NbO_x memristors. **h**, Transfer characteristics at a source-to-drain voltage (V_{DS}) of 3V and measured in the dark for 100 In₂O₃ phototransistors with channel lengths (L) of 10 μm and channel widths (W) of 400 μm. **i-l**, Device-to-device variation is represented using histograms of saturation mobility values (μ_{sat}) (**i**), threshold voltage (V_{th}) (**j**), subthreshold slopes (SS) (**k**) and current on/off ratios (I_{on/off}) (**l**).

We added the following statements to the revised manuscript (paragraph 1 and 2, page 6, highlighted): “*The extracted coefficient of variation (C_v) values of different parameters (V_{th}, V_{hold}, and V_{forming}) of 100 NbO_x Mott memristors are 0.0349, 0.0303, and 0.0233, respectively, demonstrating low device-to-device variability (Supplementary Fig. 7).*” “*The corresponding device-to-device variation in μ_{sat}, V_{th}, SS, and I_{on/off} of 100 In₂O₃ phototransistors with values of 0.19, 0.36, 0.21, and 0.52, respectively demonstrate a low variability (Supplementary Fig. 7).*”

6. It is also not clear whether both devices are fabricated on the same chip or externally connected. I will be more excited if the two devices are integrated on the same chip.

Response: We appreciate your interest in the device integration aspect of our research.

In our study, both the NbO_x Mott memristor and In₂O₃ phototransistor could be integrated on the same chip, as illustrated in Figure R6. The fabrication is compatible with conventional microfabrication process. The electrical performance of the monolithically integrated devices was illustrated in Figure R3.

The monolithic integration could further improve scalability and functionality. Also, large arrays could be designed and fabricated for more complicated tasks in future works.

We thank the suggestion, and we add Figure R6 as Supplementary Fig. 12 and corresponding descriptions in the revised manuscript (paragraph 1, page 9) in the revised manuscript.

Figure R6. **a**, Optical image of compactly integrated In_2O_3 optoelectronic synaptic transistors and NbO_x Mott neurons (1T1R). **b**, The flow chart of 1T1R cell preparation.

7. Which tool was used to measure the switching characteristics of NbO_x Mott memristors? As far as I know, most advanced parameter analyzers have resolutions of 10s of nanoseconds. How did the authors make such precise measurements?

Response: We utilized the waveform generator/fast measurement unit module of the Agilent B1500 semiconductor parameter analyzer integrated with a probe station system (Summit1100B) for our measurements, which provides a minimum time resolution of 10 nanoseconds.

As shown in Figure R7, the off/on or on/off switching time is in several tens of ns range. To be more accurate, we have revised Figure R7b (Supplementary Fig. 6 in the manuscript) by depicting all the measured data points. The extracted off/on switching time is <40 ns from HRS to LRS (at current of 4 mA) (left panel), and the on/off switching time is <50 ns from LRS to HRS (right panel).

Figure R7. Transient switching response of the NbO_x memristor. a, The programmed voltage-pulse input (blue curve) and corresponding current response (red curve). **b,** The switching speed is < 40 ns from off-state to on-state (left) and is < 50 ns from on-state to off-state (right).

References

1. Zhang X., et al. An artificial spiking afferent nerve based on mott memristors for neurorobotics. *Nat. Commun.* **11**, 51 (2020).
2. Han J. K., Yun S. Y., Lee S. W., Yu J. M. & Choi Y. K. A review of artificial spiking neuron devices for neural processing and sensing. *Adv. Funct. Mater.* **32**, (2022).
3. Zhu J., et al. A heterogeneously integrated spiking neuron array for multimode-fused perception and object classification. *Adv. Mater.* **34**, e2200481 (2022).
4. Yuan R., et al. A calibratable sensory neuron based on epitaxial VO₂ for spike-based neuromorphic multisensory system. *Nat. Commun.* **13**, 3973 (2022).
5. Wang X., et al. Vertically integrated spiking cone photoreceptor arrays for color perception. *Nat. Commun.* **14**, 3444 (2023).
6. Subbulakshmi Radhakrishnan S., et al. A sparse and spike-timing-based adaptive photoencoder for augmenting machine vision for spiking neural networks. *Adv. Mater.* **34**, e2202535 (2022).
7. Wu Q., et al. Spike encoding with optic sensory neurons enable a pulse coupled neural network for ultraviolet image segmentation. *Nano Lett.* **20**, 8015-8023 (2020).
8. Wang F., et al. A two-dimensional mid-infrared optoelectronic retina enabling simultaneous perception and encoding. *Nat. Commun.* **14**, 1938 (2023).
9. Kim S., et al. Artificial stimulus-response system capable of conscious response. *Sci. Adv.* **7**, (2021).
10. Han J. K., et al. Bioinspired photoresponsive single transistor neuron for a neuromorphic visual system. *Nano Lett.* **20**, 8781-8788 (2020).
11. Chen C., et al. A photoelectric spiking neuron for visual depth perception. *Adv. Mater.* **34**, e2201895 (2022).
12. Pei Y., et al. Artificial visual perception nervous system based on low-dimensional material photoelectric memristors. *ACS Nano* **15**, 17319-17326 (2021).
13. Kumar S., Williams R. S. & Wang Z. Third-order nanocircuit elements for neuromorphic engineering. *Nature* **585**, 518-523 (2020).
14. Jiang T., et al. Tetrachromatic vision-inspired neuromorphic sensors with ultraweak ultraviolet detection. *Nat. Commun.* **14**, 2281 (2023).

REVIEWER COMMENTS

Reviewer #1 (Remarks to the Author):

The manuscript lacks innovation and in-depth research, and even after revision, it still does not meet my expectations. I believe it is not suitable for publication in Nature Communications and suggest submitting it to another journal.

Reviewer #2 (Remarks to the Author):

The authors have addressed my comments to satisfaction. I recommend the publication of this manuscript in Nature Communications.

Reviewer #3 (Remarks to the Author):

See attached.

The authors have addressed most of the comments proposed by the reviewers. However, several key points are still confusing.

Comments from reviewer 1:

1) It's not uncommon to see the combination of RRAM devices and TS devices to implement LIF neuron functionality. The same group also did similar work published on IEDM 2023, the innovation should be further claimed.

2) The concept of rate and timing fusion coding is interesting, it is at an algorithm level. However, the algorithm is still not clearly demonstrated in the paper. I think adding a more detailed description of the algorithms and supplying the comparison with rate and timing coding based on Mott neurons is exciting. I suggest the authors address this question according to the following questions to further polish the manuscript.

1. The explanation of how the fusion of the two encoding methods, i.e. RTF, is utilized in the article is not clear. If fusion encoding is used, how is the network's output decoded? To be specific, how is the predicted value in Method eq.3. decoded in the TTFS and RTF scheme respectively? I guess that only the input spike trains of three kinds of coding are changed but the BP process and the decoding scheme are following rate-coding. So it is unfair to compare the loss of rate-coding, TTFS-coding, and RTF coding in the same figure.

2. The training of TTFS-coding SNN is more difficult than rate-coding because $\frac{\partial L}{\partial w^l} =$

$\frac{\partial L}{\partial t^l} \frac{\partial t^l}{\partial v^l} \frac{\partial v^l}{\partial w^l}$, in which the surrogate gradient of $\frac{\partial t^l}{\partial v^l}$ is harder to define. So the TTFS-coding SNN has a performance gap with rate-coding SNN. In comparison, in the rate coding scheme, as Method eq.4. 5. depict, many functions can be candidates for the surrogate gradient. A deep TTFS-coding SNN model with a structure of resnet-18 is difficult to converge, so the algorithm works of totally TTFS-coding deep SNN are rare. Therefore, declaring how the TTFS-coding and RTF-coding SNN are trained is very important because it directly determines whether the results are convincing. Please attach references of the algorithm if it is essential.

Responses to NCOMMS-23-46955A-Z

We appreciate all reviewers' constructive comments and valuable suggestions. We have carefully read through the reviewers' comments and thoroughly addressed all the requests and concerns. And hopefully, our responses can release your concerns. All the modifications are highlighted in the manuscript and Supplementary document. Reviewer comments are numbered in black, and our responses are in blue. Following are our point-by-point responses:

Response to Reviewers

Reviewer #1:

The manuscript lacks innovation and in-depth research, and even after revision, it still does not meet my expectations. I believe it is not suitable for publication in Nature Communications and suggest submitting it to another journal.

Response: Emulating the biological visual sensory transduction process, our work first demonstrates an artificial neuron device capable of directly encoding visual analog signals into spike trains using multiplexed RTF encoding scheme. It is the first experimental demonstration of both rate and time-to-first spike (TTFS) coding at the device level. The RTF encoding scheme synergizes the merits and addresses the limitations of both rate and TTFS coding schemes. These schemes are celebrated for their remarkable efficiency and their fidelity to biological processes, particularly in processing visual data. In comparison, conventional methods for implementing neuronal encoding functions in SNN have mainly depended on software algorithms or digital circuits. Although these approaches are effective, they require more computational power and energy.

Our work marks a pivotal leap forward by showcasing a comprehensive experimental demonstration of seamlessly integrating multiplexed rate and TTFS coding within a single device.

Although we cannot easily agree on your feedback about the lack of innovation and depth in our paper, we appreciate your comments and will continue to explore more ways to deepen our research in future works.

Reviewers 2:

The authors have addressed my comments to satisfaction. I recommend the publication of this manuscript in Nature Communications.

Response: Thank you for your comments that have enabled us to provide a much-improved manuscript.

Reviewer 3:

The authors have addressed most of the comments proposed by the reviewers. However, several key points are still confusing.

Response: Thank you for your comments. We have resolved your concerns in the revised manuscript as described below.

Comments from reviewer 1:

1) It's not uncommon to see the combination of RRAM devices and TS devices to implement LIF neuron functionality. The same group also did similar work published on IEDM 2023, the innovation should be further claimed.

Response: We thank the reviewer for bringing up this point. We highly appreciate the great efforts that the reviewer has devoted to our manuscript and the essential interest that the reviewer has shown towards our work presented at IEDM 2023. Our IEDM work only employed frequency encoding (rate coding) via the 1PT1TS structure for sparse sampling by leveraging the switching characteristics of a transistor. That work primarily utilized frequency coding, a foundational algorithm for SNN, but did not explore TTFS coding or multiplexed rate and TTFS coding. We have included our IEDM publication in the reference list as Ref. 23 of the revised manuscript.

In addition, there are several reports combining the RRAM and TS devices (e.g., metal filaments based TS) to implement LIF neuron functionality. However, it was hard to achieve high-order neural dynamics in these devices. And the implementation of multiplexed rate and TTFS coding has yet been demonstrated at the hardware level.

In this work, we observed both the rate and TTFS behaviors in the artificial neuron device, which enabled the hardware implementation of the multiplexed rate and TTFS coding (RTF coding) schemes. Our work highlights the integration of rate coding and TTFS coding within RTF coding, showcasing the robustness of rate coding in mitigating noise amidst significant input changes and utilizing the precision and inherent sensitivity of TTFS coding to minor input fluctuations. This capability enhances the artificial neuron's ability to rapidly and accurately perceive in complex environments, showcasing the significant effectiveness of hardware encoder for SNN.

2) The concept of rate and timing fusion coding is interesting, it is at an algorithm level.

However, the algorithm is still not clearly demonstrated in the paper. I think adding a more detailed description of the algorithms and supplying the comparison with rate and timing coding based on Mott neurons is exciting. I suggest the authors address this question according to the following questions to further polish the manuscript.

Response: Thank you for your insightful comments. We have provided additional details in terms of encoding methods and device characteristics in the revised manuscript as indicated below.

Q1a: The explanation of how the fusion of the two encoding methods, i.e. RTF utilized in the article is not clear.

Response: We appreciate this insightful suggestion. In this paper, we introduce three different coding techniques to convert the original image into spike trains, which are used as the input signals for the SNN model. To balance the computational complexity and prediction accuracy, we select spike train with a time-steps length of 15 to train the SNN model. It is worth noting that the parameter of time step is 1 μ s, and the length of 15 is the choice after multiple optimizations. The three coding techniques are TTFS coding, rate coding, and RTF coding. Detailed explanations of these encoding methods will be provided in the sections that follow:

(1) Rate coding technique

Our proposed device exhibits a tunable spike frequency response characteristic under varying light intensities. The relationship between the spike frequency and light intensity can be modeled through a linear function, based on data recorded from our fabricated device, as depicted in Figure R1a. The fitting function is outlined as follows:

$$y = 0.1125 + 0.425 x$$

where y and x represent spike frequency and light intensity, respectively.

Then, we utilize the same linear fitting function to obtain the spike counts on different pixel grayscale, as shown in Figure R1b. The pixel grayscale also ranges from 0 to 255, and the spike counts is in the range of 1 to 15. The obtain spike frequency is float number, which need to conduct floor operation to get an integer number. Finally, we generate spike trains according to this spike frequency where the spike number and spike time follows Poisson distribution.

Figure R1. Spike frequency as a linear function of the light intensity. **a**, The artificial visual spiking neuron exhibits a tunable spike frequency response characteristic under varying light intensities. **b**, Mapping function depicting the relationship between spike counts and varying pixel grayscale. The red line indicates the fitted curve parameters, which was utilized in the mapping algorithm.

To make it clear, we've listed the specific rate coding steps below:

Algorithm 1: Rate coding

- 1: Fitting the relationship between spike frequency (MHz) and light intensity where the data is recorded from the fabricated device;
- 2: Based on the same fitting function, calculate the spike frequency (time step) under different pixel grayscale of the original image;
- 3: Conduct a floor operation to obtain spike frequency (integer number);
- 4: Generate a spike train according to this spike frequency where the spike frequency and spike time follow a Poisson distribution.

(2) TTFS coding technique

Our proposed device exhibits a first-spike latency characteristic under varying light intensities. The relationship between the first spike latency and light intensity can be well fitted by an exponential decay function, as shown in Figure R2a. The fitting function is listed as follows:

$$T_{latency} = 1.01 + 675 \exp(-3.02 x)$$

where $T_{latency}$ represents the TTFS, and x represents the light intensity.

Then, we utilize the exponential decay function to obtain the first spike latency on different pixel grayscale, as shown in Figure R2b. The pixel grayscale ranges from 0 to 255, and the first spike latency is in the range of 15 to 1. The obtain first spike latency is float number, which need to conducte floor operation to get an integer number.

Finally, we generate spike trains where the time of this only spike is the first spike latency (integer number).

Figure R2. First spike latency time as an exponential function of the light intensity. **a**, The artificial visual spiking neuron exhibits a tunable first spike latency response characteristic under varying light intensities. **b**, Mapping function depicting the relationship between first spike latency and varying pixel grayscale. The red line indicates the fitted exponential decay function, which was utilized in the mapping algorithm.

To make it clear, we've listed the specific TTFS coding steps below:

Algorithm 2: TTFS coding

- 1: Fitting the relationship between first spike latency (μs) and light intensity where the data is recorded from the fabricated device;
- 2: Based on the same fitting function, calculate the first spike latency (time step) under different pixel grayscale of the original image;
- 3: Conduct a floor operation to obtain the first spike latency (integer number);
- 4: Generate a spike train where the time of this only spike is the first spike latency (integer number).

(3) Rate and temporal fusion (RTF) coding

The RTF coding incorporates the above two features, where the spike frequency (F_{Rate}) and first spike latency are obtained using the same fitting function with rate and TTFS coding techniques, respectively. Then, we need to calculate the number of time steps (F_{rest}) remaining after first spike in TTFS coding. If the F_{rest} is larger than F_{Rate} , the $F_{Final} = F_{Rate}$. Otherwise, $F_{Final} = F_{rest}$. We first ensure the location of the first spike using the result of TTFS coding. The rest of the spike train depends on the F_{Final} . If $F_{Final} = F_{Rate}$, the rest of spike train is generated according to the spike frequency from rate coding. If $F_{Final} = F_{rest}$, then every position after the first

spike will have one spike.

Algorithm 3: RTF coding

- 1: Obtaining the first spike latency (time step) and spike frequency with the same function of TTFS and rate coding;
- 2: Calculating the number of time steps (F_{rest}) after first spike in TTFS coding;
- 3: Comparing the F_{rest} and F_{Rate} to obtain the final frequency F_{Final} ;
- 4: If $F_{Final} = F_{Rate}$, the rest of spike train is generated according to the spike frequency from rate coding;
- 5: If $F_{Final} = F_{rest}$, then all position after first spike will have one spike.

Figure R3. Spike raster of a self-driving recording image for rate coding, TTFS coding, and RTF coding in 15 time-steps. Spike trains of different coding methods for image locations along the column are marked by arrows. The spike sequence depicted within the red dashed box is derived from the pixels enclosed by the red box. In rate coding, there exists a linear relationship between the pixel grayscale value and the frequency of the resulting spike train. Conversely, in TTFS coding, the pixel grayscale is connected to the first spike latency via an exponential decay function. The generated spike train in RTF coding incorporates these two fitting relationships.

Finally, we utilize one original image to demonstrate all these three coding techniques, as shown in Figure R3. The pixel grayscale has a linear relationship with the frequency of the generated spike train in rate coding. The pixel grayscale is related to the first spike latency in TTFS coding through an exponential decay function. The

generated spike train in RTF coding incorporates these two fitting relationships.

We added the following statements to the revised manuscript (page 12, highlighted): “*These three coding schemes are detailed in Supplementary Note 1-3.*” Additionally, we have added corresponding discussion in the revised manuscript (Please see *Supplementary Fig. 18,19* and *Supplementary Note 1-3*). Moreover, we have updated the *Supplementary Fig. 20* in the revised version of the manuscript.

Q1b: *If fusion encoding is used, how is the network's output decoded? To be specific, how is the predicted value in Method eq.3. decoded in the TTFS and RTF scheme respectively?*

Response: Thanks for the excellent comments. When we utilize the SNN in the prediction task, it is composed of three parts: the input spike trains generated by different coding schemes, the SNN model, and the predicted value, as shown in Figure R4 (Figure 5a in the manuscript). The decoding process relies on extracting the spatial and temporal features of the input spike sequences.

In encoding the original image into spike trains, the TTFS coding utilizes the first spike latency to represent the pixel grayscale. Because rate-coding utilizes the spike frequency to represent the pixel grayscale, rate and RTF coding has the two temporal characteristics mentioned above. Furthermore, each pixel of the original image is encoded into spike trains with TTFS and RTF coding techniques. Thus, the input spike trains embody the spatial feature of the original scene. Overall, the generated spike trains have spatial and temporal features. The difference is that RTF coding has spike frequency and first spike latency characteristics in the temporal domain, while the TTFS coding primarily features first spike latency.

These encoded sequences are processed by an SNN model utilizing a ResNet-18 architecture composed of 17 spike layers and one fully connected layer. Each spike layer includes one convolution layer and Leaky integrate-and-fire (LIF) neurons. The convolution layer is responsible for extracting spatial features from input spike trains, while the LIF neuron is designed to extract temporal feature from input spike trains. The SNN model can predict a car's steering angle and speed based on the extracted spatial and temporal features. The predicted value depends on different prediction tasks, such as steering angle or speed, and will be directly readout without additional decoding.

We have updated the *Figure 5a* in the revised version of the manuscript. Now,

Figure 5a accurately depicts the distinction between rate and TTFS coding schemes.

Figure R4. The prediction tasks comprising three main components: the input spike trains generated by different coding schemes, the SNN model, and the output predicted value.

Q1c: *I guess that only the input spike trains of three kinds of coding are changed but the BP process and the decoding scheme are following rate-coding. So it is unfair to compare the loss of rate-coding, TTFS-coding, and RTF coding in the same figure.*

Response: Thanks for raising this question. The BP process and decoding scheme belong to the part of the spiking neural network. When we adopt different coding techniques in the predicted task, we retrain the SNN model and adjust the hyperparameters to get the best results with the corresponding input spike trains generated in different encoding methods. Thus, all the three encoding inputs have been trained to the optimal, and we believe the comparison among them is sound.

Q2a: *The training of TTFS-coding SNN is more difficult than rate-coding because $\frac{\partial L}{\partial w^l}$ = $\frac{\partial L}{\partial o^l} \frac{\partial o^l}{\partial v^l} \frac{\partial v^l}{\partial w^l}$ in which the surrogate gradient of $\frac{\partial o^l}{\partial v^l}$ is harder to define. In comparison, in the rate coding scheme, as Method eq.4. 5. depict, many functions can be candidates for the surrogate gradient.*

Response: Thank you for raising this interesting and inspiring question. As mentioned in the Q1, the surrogate function of spiking neural network in the backpropagation process belongs to SNN part, thus the problem about the surrogate gradient of $\frac{\partial o^l}{\partial v^l}$ is

the same for all these three coding techniques. The relationship between o^l and V^l is listed as follows:

$$o^l = \begin{cases} 1, & \text{if } V^l > V_{th} \\ 0, & \text{if } V^l \leq V_{th} \end{cases}, \quad (1)$$

where the V_{th} is the threshold of LIF neuron whether to fire a spike. This function is a step function, it is not differentiable in the process of backpropagation. As you mentioned, there are many functions that can be candidates for the surrogate gradient, which could be found in Figure R5^[R2].

Figure R5. (a) Several surrogate function candidates and (b) their derivative for step function. The figure is taken from Ref. R2.

(1) Sigmoid function:

$$f(x) = \frac{1}{1 + e^{-\alpha x}}, \quad (2)$$

(2) Arctan function:

$$f(x) = \frac{1}{\pi} \arctan\left(\frac{\pi}{2} \alpha x\right) + \frac{1}{2}, \quad (3)$$

(3) Softsign function:

$$f(x) = \frac{1}{2} \left(\frac{\alpha x}{1 + |\alpha x|} + 1 \right), \quad (4)$$

(4) Erf function:

$$f(x) = \frac{1}{\sqrt{\pi}} \int_{-\infty}^{\alpha x} e^{-t^2} dt, \quad (5)$$

The best fitting performance is the erf function, but the calculation process is too complex that is not suitable for spiking neural network model. To balance the performance and computational complexity, we select softsign function in our proposed spiking neural network model, which is used for all these three coding techniques.

More information about surrogate gradient learning in spiking neural networks could be found in references [Ref. R1-3], which were cited as Ref. 47-49 in the revised manuscript.

Q2b: The TTFS coding SNN has a performance gap with rate-coding SNN. A deep TTFS-coding SNN model with a structure of ResNet-18 is difficult to converge, so the algorithm works of totally TTFS-coding deep SNN are rare.

Response: Thanks for the comments. As mentioned above, the performance of SNN depends on two factors: the input spike trains encoded by specific encoding schemes and the feature extraction of the SNN model. These coding schemes have spatial and temporal features. The proposed SNN model adopts a ResNet-18 structure to extract the above spatial and temporal features in input spike trains. ResNet-18 is composed of 17 spike layers and one fully connected layer. Each spike layer includes one convolution layer and LIF neurons. The convolution layer is responsible for extracting the spatial feature from input spike trains, which can extract spatial feature from input spike trains of both coding schemes. LIF neuron extracts the temporal feature from input spike trains. Since LIF neurons extract the temporal characteristics of the input spike train through the accumulation and leakage of membrane potentials, the temporal feature extracting process usually requires the interaction of multiple spikes. Because TTFS encoding only has one spike, this is why there is a performance gap between rate-coding and TTFS coding.

During the training process, the spiking neural network model has the ability to adapt to different input types. For the input spike trains encoded by TTFS method, the model increases the connectivity weights of presynaptic neurons, allowing the firing of LIF neurons even under only one input spike. As shown in Figure R6, the spike firing rate of TTFS and rate coding in steering angle prediction task is 0.12 and 0.17, respectively. The spike firing rate of TTFS and rate coding in speed prediction task is 0.13 and 0.17, respectively. Thus, the spike firing rate of TTFS is only slightly lower than that of rate coding in the spiking neural network, which also supports feature extraction and prediction of the steering angle and speed of a car.

Figure R6. a, Comparison of three coding schemes in terms of the spike firing rate and loss in predicting the steering angle. **b**, Comparison of the spike firing rate and loss in the speed prediction task.

Figure R7. a, Loss value per epoch during training of the SNN for steering angle prediction under different coding schemes. **b**, Loss value per epoch during training of the SNN for speed prediction under different coding schemes.

As shown in Figure R7, the SNN model with the input spike trains from TTFS and rate coding can converge after 45 epoch. However, the pace of convergence in TTFS coding is slower than in rate coding.

Q2c: Therefore, declaring how the TTFS-coding and RTF-coding SNN are trained is very important because it directly determines whether the results are convincing. Please attach references of the algorithm if it is essential.

Response: Thanks for the constructive suggestion. The training process of SNN with TTFS-coding and RTF-coding can be divided into three parts: the input spike trains and

labels, SNN model, and the hyperparameter for training the model.

TTFS-coding and RTF-coding are used to convert the pixel grayscale to spike train, and the length of generated spike train by two coding techniques is 15 time steps. Meanwhile, the resolution of original image is 100×140 . Thus, the converted data format of these two techniques for this original image is $15 \times 100 \times 140$, which serves as the input data for SNN. The label for SNN is steering angle or speed value, which is independent of the coding technique, but depends on the different prediction tasks.

Figure R8. The Leaky-Integrate-and-Fire (LIF) neuron model of SNN.

The proposed SNN model adopts a ResNet structure, which is composed of 17 spike layers and one fully connected layer. Each spike layer consists of one convolution layer and LIF neurons. In the forward computation process, the input spike signal will pass through these layers in turn, conducting spatial and temporal extraction and final prediction. The working mechanism of this convolution layer is the same as the traditional convolution layer. The distinction lies in the convolution layer of the proposed SNN model receiving spike signals as its input, and the traditional convolution layer input is floating-point numbers. The working mechanism of LIF neurons is summarized in Figure R8. The updated membrane potential is obtained by input spike multiplying by the weight value. This updated potential will be added to the existing membrane potential of LIF neuron, then compared with threshold to decide whether to fire a spike. The new firing spike will be transferred to conduct the subsequent computation in the next spike layer.

After the final computation of the fully connected layer, the SNN model will calculate the error between the predicted values and labels. Then, the model will perform an error backpropagation process. For a non-differentiable function, like step

function in LIF neuron, the model adopts a surrogate gradient function to replace it, and completes the error backpropagation.

The whole model is built with a SNN model library – spikingjelly, which is widely used in various tasks with an SNN model. The link to spikingjelly is listed here, <https://github.com/fangwei123456/spikingjelly/tree/master>. The framework is implemented on an NVIDIA RTX 3060.

We take the steering angle prediction task as an example to introduce the hyperparameter of SNN model in the training process. For the rate coding spike train, we use an Adam optimizer with a base learning rate of $1e-4$, β_1 , $\beta_2=0.9, 0.999$, and a weight decay of $1e-6$ to minimize the joint loss. For the input spike train with TTFS coding, the base learning rate is increased to $3e-4$ for obtaining suitable parameters over a larger range. Then, LIF neurons can fire a spike to post-synapse neuron even with very few spike input. The weight decay is also increased to $4e-3$. Because the increase in the learning rate can easily lead to model overfitting, it is necessary to increase weight decay to give a certain penalty to the model parameters and solve the problem of model overfitting.

We added several more references about training SNN model with input spike trains generated by TTFS and rate coding [Ref. R4-6], which were added as Ref. 50-52 in the revised manuscript.

References:

- [R1] Neftci, E. O., Mostafa, H. & Zenke, F. Surrogate gradient learning in spiking neural networks: Bringing the power of gradient-based optimization to spiking neural networks. *IEEE Signal Process. Mag.* 36, 51-63 (2019).
- [R2] Xiang, S., Jiang, S., Liu, X., Zhang, T. & Yu, L. Spiking VGG7: Deep convolutional spiking neural network with direct training for object recognition. *Electronics* 11, 2097 (2022).
- [R3] Li, Y., et al. Differentiable spike: Rethinking gradient-descent for training spiking neural networks. In *2021 Advances in Neural Information Processing Systems (NeurIPS)* 23426–23439 (2021).
- [R4] Sakemi, Y., Yamamoto, K., Hosomi, T. & Aihara, K. Sparse-firing regularization methods for spiking neural networks with time-to-first-spike coding. *Sci. Rep.* 13, 22897 (2023).
- [R5] Nomura, O., Sakemi, Y., Hosomi, T. & Morie, T. Robustness of spiking neural networks based on time-to-first-spike encoding against adversarial attacks. *IEEE Trans. Circuits Syst. II, Express Briefs* 69, 3640-3644 (2022).
- [R6] Guo, W., Fouda, M. E., Eltawil, A. M. & Salama, K. N. Neural coding in spiking neural networks: A comparative study for robust neuromorphic systems. *Front. Neurosci.* 15, 638474 (2021).